# Programming actuation onset of a liquid crystalline elastomer via isomerization of network topology

Guancong Chen[1,2], Haijun Feng[1], Xiaorui Zhou[1], Feng Gao[3], Kai Zhou[1], Youju Huang [2], Binjie Jin [1] ✉, Tao Xie [1] & Qian Zhao [1] ✉

Tuning actuation temperatures of liquid crystalline elastomers (LCEs) achieves control of their actuation onsets, which is generally accomplished in the synthesis step and cannot be altered afterward. Multiple actuation onsets in one LCE can be encoded if the post-synthesis regulation of actuation temperature can be spatiotemporally achieved. This would allow realizing a logical time-evolution of actuation, desired for future soft robots. Nevertheless, this task is challenging given the additional need to ensure mesogen alignment required for actuation. We achieved this goal with a topology isomerizable network (TIN) of LCE containing aromatic and aliphatic esters in the mesogenic and amorphous phases, respectively. These two ester bonds can be distinctly activated for transesterification. The homolytic bond exchange between aliphatic esters allows mechanically induced mesogen alignment without affecting the mesogenic phase. Most importantly, the heterolytic exchange between aromatic and aliphatic esters changes the actuation temperature under different conditions. Spatial control of the two mechanisms via a photo-latent catalyst unleashes the freedom in regulating actuation temperature distribution, yielding unusual controllability in actuation geometries and logical sequence. Our principle is generally applicable to common LCEs containing both aromatic and aliphatic esters.

Monodomain liquid crystalline elastomers (LCEs) perform muscle-like actuation in response to external stimulation, making them attractive for soft robotics and medical devices[1–3]. Thermotropic LCEs rely on heating above or cooling below the nematic-to-isotropic temperature ($T_{NI}$) for actuation[4,5]. Past progresses have focused mainly on new approaches for mesogen alignment[6–9], enhancing actuation strain[10,11], and increasing work capacity[12–14]. Of equal importance is the manipulation of $T_{NI}$ to control the actuation onset of LCEs, which has received less attention. Mechanistically speaking, the former indicates the attainable functions of a soft actuator and the latter defines its

applicable application scenarios. For instance, the $T_{NI}$ of LCEs for biomedical applications typically shall be lowered to 70 °C[4]. Manipulating the $T_{NI}$ hereafter paves the way to make LCEs functional in distinct conditions. Beyond this, sequential actuation, indicating higher controllability in shape-changing, can be realized via encoding multiple $T_{NI}$s into one LCE, which would allow us to explore new applications of LCE-based devices.

Driven by this, two main approaches have been developed to tune $T_{NI}$: engineering the molecular structure of mesogens and optimizing the network topology. For the former, adjusting the π−π interaction in

[1]State Key Laboratory of Chemical Engineering, College of Chemical and Biological Engineering, Zhejiang University, Hangzhou 310027 Zhejiang, China. [2]College of Material, Chemistry and Chemical Engineering, Key Laboratory of Organosilicon Chemistry and Material Technology, Ministry of Education, Hangzhou Normal University, Hangzhou 311121 Zhejiang, China. [3]National Engineering Laboratory for Textile Fiber Materials & Processing Technology, Zhejiang Sci-Tech University, Hangzhou 310018 Zhejiang, China. ✉e-mail: binjie_jin@zju.edu.cn; qianzhao@zju.edu.cn

the mesogen core[15] and the length of the aliphatic tails[16] is effective. For the latter, this is realized with different chain extenders to regulate the crosslinking density[17] or co-polymerization of nonliquid crystalline components to control the overall liquid crystalline content[18,19]. However, in these aforementioned cases, the $T_{NI}$ is predetermined by the formulation and would not change after the network formation. To achieve a different $T_{NI}$, a new sample must be fabricated.

The above correlation between $T_{NI}$ and the network structure reminds us of the recently emerged topology isomerizable network (TIN)[20,21]. For a typical dynamic covalent network, the topology is consistent before and after the bond exchange[8,22]. In contrast, TIN can realize the designable topological transformation via bond exchange, for instance, from a grafting network to a brush network. Thus, a single TIN can be programmed into distinct materials with different properties (e.g., modulus and crystallinity) whenever required. An intriguing question arises: can a dynamic covalent LCE network be designed to isomerize its topology as a way of regulating the $T_{NI}$ on demand? This issue is still pending. Although dynamic covalent LCE networks have been extensively studied, none presents the desired capability[8,22,23]. Typically, reversible actuations are enabled after the mesogen alignment is fixed by bond exchange under external force. However, their $T_{NI}$s remain unchanged since the network topologies are not isomerized. We should note that Yao et al. recently reported that annealing a dynamic covalent LCE network can alter the $T_{NI}$, which was attributed to the change in the structural order of the mesogenic phase[24]. Specifically, annealing at $T_a < T_{NI}$ encourages forming a more compact layered structure corresponding to a higher structural order and higher $T_{NI}$. In contrast, annealing at $T_a > T_{NI}$ had an opposite effect, that is, it impaired the regularity of the lamellar stacking and led to a lower $T_{NI}$. The change in structural order and $T_{NI}$ can be fixed by quenching the bond exchange reaction. Despite the elegance, this post-modification of $T_{NI}$ required a long annealing time (usually days) and the chemical topologies was not altered in the process. It should be emphasized that adopting the concept of TIN into LCEs is a challenging task, as the mesogen alignment should be well maintained during the topology isomerization for reversible actuation. This implies the need for designing two sets of dynamic bond exchanges in one LCE network for mesogen alignment and topology isomerization.

In this work, we accomplish this goal with a dynamic LCE network containing aromatic and aliphatic esters in its mesogenic and amorphous phases, respectively. The transesterification of these two ester bonds can be distinctly triggered under different conditions. As such, the homolytic bond exchange between aliphatic esters induces network arrangement for mesogen alignment. The heterolytic exchange between aromatic and aliphatic esters is activated to alter the mesogenic structure (network isomerization) for programming $T_{NI}$. Building upon this, we further illustrate that the two bond exchange mechanisms can be spatial-selectively controlled via a photo-latent catalyst. This leads to pixelated programming of both the alignment and $T_{NI}$, going beyond prior work that only allows pixelated mesogen alignment. The material design principle behind our dynamic covalent LCE results in unusual versatility for future LCE-based soft robots.

## Results

### Network design and programming mechanisms

The topology-transformable LCE network was fabricated based on the reaction between 1,4-bis-[4-(6-acryloyloxy-hexyloxy) benzoyloxy]−2-methylbenzene (RM82, LC mesogen) and 2,2′-(ethylene-dioxyrl)-ethanethiol (EDDT, chain extender), initiated by 2,2-dimethoxy-2-phenyl-acetophenone (DMPA, photo-initiator) (Fig. 1a)[25]. The molarities of RM82 and EDDT are maintained equally. Upon UV exposure, the thiol-Michael addition (between EDDT and RM82) and the homo-polymerization of acrylate groups were simultaneously initiated by the generated free radicals (Supplementary Fig. 1)[26]. The overall result is that a crosslinked LCE network was obtained after UV irradiation,

where the aliphatic and aromatic esters are located within the amorphous and mesogenic phases, respectively (Fig. 1b).

Upon heating, two distinct transesterification reactions can be activated. One is the bond exchange between the aliphatic ester groups, referred to as the homolytic reaction. The other one is the bond exchange between the aliphatic and aromatic ester groups, referred to as the heterolytic reaction. As illustrated in Fig. 1c, the mesogen cores are preserved during the homolytic reaction while partially altered by the heterolytic one. Given the bond exchange nature, the alignment programming can be realized via a thermal-mechanical approach (Fig. 1d)[8,13,22]. Specifically, when a polydomain LCE is uniaxially stretched, the mesogens are aligned along the stretching direction. Upon heating, the dynamic network topology transformations are enabled due to the activated bond exchange reactions. The stretching-induced alignment is consequently fixed after cooling down, which can be translated into reversible actuation upon temperature change.

However, things are different when comparing the liquid crystalline content of the original and programmed networks. We note here that the liquid crystalline content refers to the amount of mesogens derived from RM82 within the network. Specifically, the liquid crystalline content is maintained when the topology transformation is enabled by the homolytic reaction. In contrast, it is reduced through the heterolytic reaction since the mesogen structures are altered. Accordingly, the former topology transformation is referred to as network rearrangement to indicate the network structure is preserved after programming. And the latter is referred to as network isomerization to emphasize the change in network topology (Fig. 1d). Thus, the $T_{NI}$ of the LCE network programmed based on network rearrangement is preserved, while the one based on network isomerization is lower than the original $T_{NI}$. As such, beyond the mesogen alignment, our topology-transformable LCE network allows us to program the $T_{NI}$ after network synthesis by selectively activating the two distinct network topology transformations. We note here that the network rearrangement and isomerization can also be conducted without external force, which allows to program the $T_{NI}$ of polydomain LCEs.

In principle, activating the heterolytic reaction within an LCE network demands a higher temperature or catalyst concentration than the homolytic one[27]. The two reactions therefore can be distinctively triggered at different conditions, which allows us to program the $T_{NI}$ by varying the programming temperature ($T_p$) and/or catalyst concentration ($C_{cat}$). Moreover, it can be spatially conducted when locally different $T_p$s and/or $C_{cat}$s are applied. Figure 1e shows two approaches for registering two different $T_{NI}$s into a single LCE network. One is varying the $T_p$ when $C_{cat}$ is fixed and the other one is varying the $C_{cat}$ when $T_p$ is fixed. As such, the heterolytic reaction within the region subjected to the high $T_p$ or $C_{cat}$ is more pronounced, which results in a lower $T_{NI}$ than that of the other region programmed with the low $T_p$ or $C_{cat}$.

Together with uniaxial stretching, a monodomain LCE with two distinct $T_{NI}$s (left part: $T_{NI\text{-low}}$, right part: $T_{NI\text{-high}}$) is obtained after programming, which can exhibit a thermally induced stepwise actuation (Fig. 1f). Upon heating, the sample's left part contracts once the temperature is above the $T_{NI\text{-low}}$, whereas the right part is activated until heated above the $T_{NI\text{-high}}$. Upon cooling, the right part recovers to its initial length when the temperature is between the $T_{NI\text{-high}}$ and $T_{NI\text{-low}}$, while the left part elongates until the temperature continuously decreases below the $T_{NI\text{-low}}$. We note here that more than two $T_{NI}$s can be embedded if desired via employing more complex profiles of patterned $T_p$ and/or $C_{cat}$.

### $T_{NI}$ programming based on temperature control

To validate the hypothesized mechanism in Fig. 1c, triazabicyclo[4.4.0] dec-5-ene (TBD) neutralized with acetic acid (1%) was introduced into

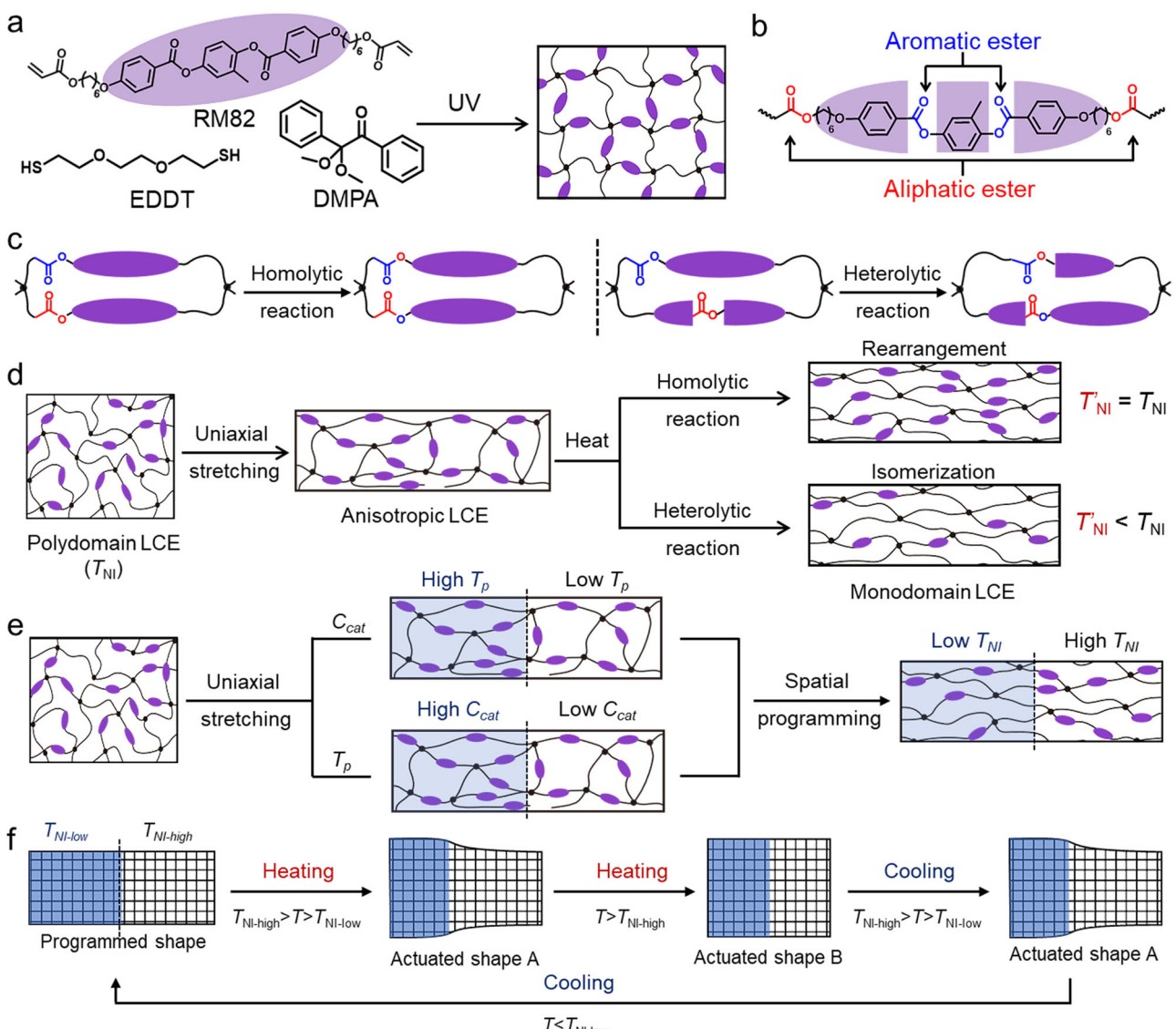

**Fig. 1 | Synthesis and programming of the topology-transformable liquid crystalline elastomer. a** Synthesis route. **b** Two sorts of esters within the network. **c** Homolytic and heterolytic bond exchange reactions. **d** Network reconfiguration and isomerization during alignment programming. **e** Spatial programming of the alignment and $T_{NI}$. **f** Sequential actuation upon progressive heating/cooling.

the LCE network as the transesterification catalyst[28]. The neutralization of TBD is to suppress its alkalinity. Otherwise, the thiol-Michael addition is initiated by it before UV irradiation, which makes the precursor solution too viscous to be handled[28]. We noted that the bonded water molecules in this hygroscopic catalyst can induce the hydrolysis of the ester bonds, generating the hydroxyl groups for transesterification[29]. Subsequently, iso-strain stress relaxation tests were conducted to evaluate the network dynamicability. The thickness of the LCE films in this paper is fixed at 0.4 mm to exclude the thermal gradient along the thickness direction. Specifically, each sample was stretched to 50% first and then annealed at a specific programming temperature ($T_p$). The time evolution of the ratio between the residual stress ($\sigma$) and the initial applied stress ($\sigma_0$), referred to as the normalized stress ($\sigma/\sigma_0$), was monitored (Fig. 2a). Clearly, the network can relax the applied stress due to the bond exchange, which was more significant when $T_p$ increased from 120 °C to 140 °C.

The verified network dynamicability allows us to program the mesogen alignment via the thermal-mechanical approach. Specifically, mesogens are aligned along the stretching direction when the LCE is uniaxially stretched. The subsequently activated bond exchange upon

heating can fix the alignment after cooling down. To verify this, an LCE strip was uniaxially stretched to 50% and annealed at 120 °C for 10 min, where the registered alignment is evidenced by the transformation of the two-dimensional (2D) wide-angle X-ray diffraction (WAXD) pattern (from an isotropic ring to two distinct arcs, Fig. 2b). The relative positions of the two arcs indicate the mesogens are aligned along the stretching direction (0° and 180°), which can be further confirmed by the polarized optical microscopy (POM) images (Supplementary Fig. 2). Since the $T_{NI}$ of the as-synthesized LCE network is around 115 °C (Supplementary Fig. 3), the actuation is further visualized by the length change (between 36 mm and 23 mm) upon temperature changes (Fig. 2c).

The alignment programming efficiency can be quantitatively evaluated by the actuation strain ($\lambda$). To investigate this, LCEs were uniaxially stretched to 50% and annealed at different $T_p$s. The reversible actuation strain is calculated as follows: $\lambda = (L_{cold} - L_{heat})/L_{heat} \times 100\%$, where $L_{cold}$ and $L_{heat}$ are the sample lengths at $-20$ °C and 120 °C, respectively. Accordingly, each curve in Fig. 2d shows that the actuation strain first increases with programming time ($t_p$) but decreases after a longer duration, which is more significant at a higher $T_p$. Further

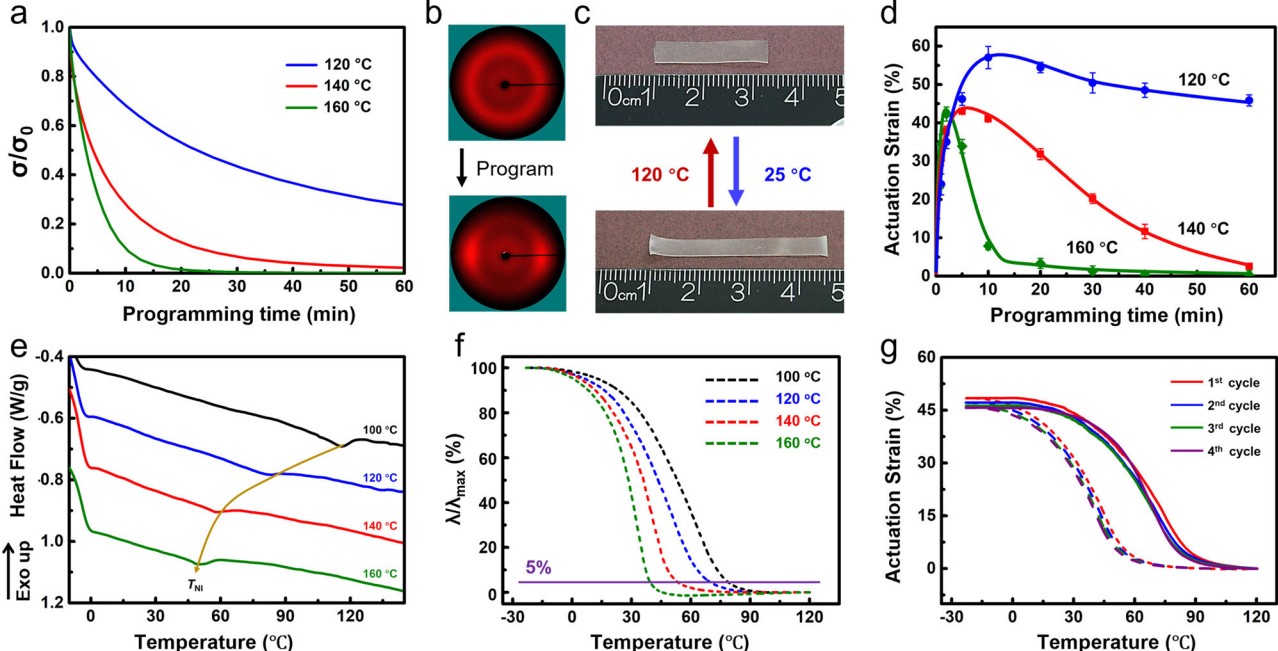

**Fig. 2 | Actuation programming of LCEs (catalyst: 1% neutralized TBD, pre-stretched strain: 50%). a** Stress relaxation under different $T_p$s. **b** Isotropic-to-anisotropic transformation of the 2D-WAXD pattern before and after alignment programming. **c** LCE linear actuation ($T_p$: 120 °C, $t_p$: 10 min). **d** Correlation between actuation strain and $t_p$ at different $T_p$s. Error bars represent standard deviation, $n = 5$. **e** $T_{NI}$s of the LCEs programmed at different $T_p$s ($t_p$: 10 min). **f** Relationship between actuation strain and temperature upon cooling ($t_p$: 10 min). **g** DMA cyclic actuation curves ($T_p$: 120 °C, $t_p$: 10 min).

comparison among the three curves reveals that the turning points are around 10, 5, and 2.5 min when $T_p$s are 120, 140, and 160 °C, respectively. And the $\lambda$ at each turning point is about 55, 43, and 41%. In principle, the drop in $\lambda$ at the later period of programming is attributed to the fact that the applied stress is already significantly relaxed, although the strain is maintained. Under such condition, the LCE network prefers to adopt an isotropic state to miniaturize the network entropy since $T_p$ is above the $T_{NI}$. The fixed alignment consequently is gradually erased and the actuation strain is reduced[30].

Beyond the alignment programming, the reduction in the liquid crystalline content that affects the $T_{NI}$ demands more attention. To investigate this, the $T_{NI}$s of the samples programmed at different $T_p$s (for 10 minutes) were investigated by the differential scanning calorimeter (DSC) analysis. The curves of the second heating run are presented in Fig. 2e, where the peak temperature is quoted as the value of $T_{NI}$. Accordingly, the $T_{NI}$s are around 82, 58, and 51 °C when the $T_p$s are 120, 140, and 160 °C, respectively. We noted here that the $T_{NI}$ of the LCE programmed at 100 is about 114 °C, which is almost identical to the original one (115 °C). This result indicates that the heterolytic reaction is rarely activated at 100 °C. The network integrity of the programmed LCEs are well maintained as proved by the considerable gel fraction (>95%, Supplementary Fig. 4).

The correlation between the actuation strain ($\lambda$) and temperature is more important when considering LCEs as actuators. In particular, the temperature when the actuation starts is a pivotal parameter to portray the shape-changing behavior of LCEs, which is referred to as actuation temperature ($T_{act}$). For a quantitative comparison, it is defined as the temperature when the normalized actuation strain ($\lambda/\lambda_{max} \times 100\%$, $\lambda_{max}$ is the total strain of one actuation cycle) reaches 5%. To investigate the relationship between $T_{act}$ and $T_p$, the normalized actuation strain of the LCEs programmed at different $T_p$s for 10 min was plotted as a temperature function in Supplementary Fig. 5. For a better visual effect, the actuation curves upon cooling were presented in Fig. 2f since they are smoother than those upon heating. Clearly, Fig. 2f shows the $T_{act}$s are 78, 68, 52, and 38 °C when the $T_p$s are 100,

120, 140, and 160 °C, respectively. We note here that the $T_{act}$ of an LCE sample deduced from the heating step is larger than the one determined from the cooling curves (Supplementary Fig. 6). This is because the sample temperature usually lags behind the applied temperature. Nevertheless, they demonstrate the same tendency that the $T_{act}$ decreases with ascending $T_p$.

Further comparison between the $T_{act}$ and $T_{NI}$ of one LCE shows their values are not the same. For instance, the $T_{act}$ is 68 °C whereas the $T_{NI}$ is 82 °C when $T_p$ is 120 °C. This difference can be attributed to the following reasons: (i) The $T_{NI}$ is determined based on the enthalpic change of a microscopic transition, while the $T_{act}$ is deduced from the temperature evolution of the macroscopic length; (ii) The $T_{NI}$ is investigated via DSC, while the $T_{act}$ is characterized via DMA. The intrinsic variance between the two analysis techniques needs to be also considered. The stability of the registered actuation strain and $T_{act}$ is further investigated. After the catalyst was removed via solvent extraction, the cyclic strain-to-temperature curves are almost identical (Fig. 2g). The above results prove that the $T_{NI}$ can be maintained or lowered during the alignment programming, depending on the selection of $T_p$. It is worth mentioning that transesterification has been adopted to prepare dynamic LCEs[8,27,31]. However, the $T_{NI}$ cannot be modified since only the homolytic bond exchange has been focused on during the alignment programming. Hereafter, we demonstrate that programming the $T_{NI}$ via selectively activating the two different transesterification mechanisms offers new opportunities to control the actuation performance for these otherwise ordinary materials.

## Spatial-temporal programming of the $T_{NI}$

The success in regulating $T_{NI}$ encourages us to consider whether a heterogeneous LCE with multiple $T_{NI}$s can be fabricated if the network isomerization is triggered spatially. Based on the thermodynamic nature, it can be reached by applying a patterned temperature field or catalyst amount. Nevertheless, the spatial pattern of the temperature is difficult to realize. We therefore resort to a photo-base generator (PBG, ketoprofen meditated TBD) to achieve the localized distribution of the

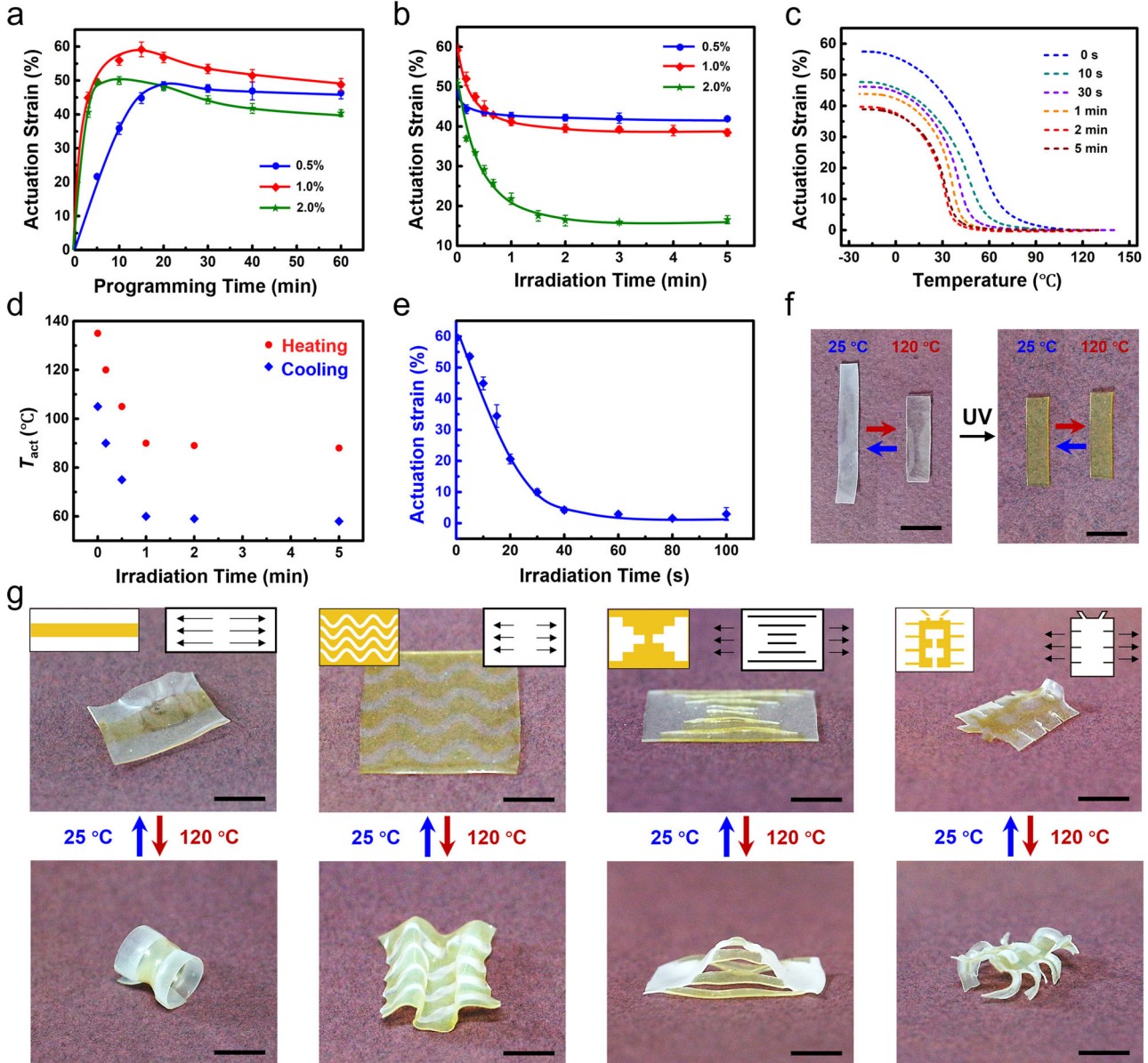

**Fig. 3 | Actuation programming of the LCEs (catalyst: PBG, pre-stretched strain: 50%, $T_p$: 120 °C). a** Dependence of the actuation strain on $t_p$ (catalyzed via unexposed PBG). Error bars represent standard deviation, $n = 5$. **b** Relationship between the actuation strain and the irradiation time ($t_p$: 10 min). Error bars represent standard deviation $n = 5$. **c** Correlation between the actuation strain and the temperature of the samples irradiated for different times ($t_p$: 10 min). **d** Relationship between $T_{act}$ and the irradiation time ($t_p$: 10 min). **e** Residual actuation strain of a monodomain LCE irradiated at 120 °C for different times (without stress). Error bars represent standard deviation, $n = 5$. **f** Visual demonstration of actuation erasing (irradiation time: 120 s). **g** 3D active LCE structures fabricated via spatially erasing actuation strain. Inset: the left one is the employed photomask, where the yellow areas are the exposed regions; the right one is the schematic illustration of the LCE sample, where the arrows indicate the alignment direction and the solid lines represent the cutting lines of the kirigami patterns. Scale bars: 1 cm.

catalyst amount via ultraviolet (UV) irradiation[20]. Specifically, TBD, the transesterification catalyst, is released from PBG upon UV irradiation (Supplementary Fig. 7), whose amount depends on the irradiation time[20]. In this case, the material could be treated as an assembly of reaction pixels with independent catalyst dosage. This corresponds to the distinct reaction degrees of the heterolytic bond exchange between the aliphatic and aromatic esters. Different $T_{NI}$s, therefore, can be locally written into the LCE network.

In principle, the loading of PBG is critical since its intrinsic alkaline (without UV irradiation) shall also be effective in the activation of transesterification. The dynamicity of the LCEs with different PBG amounts, loaded via the swelling-drying method[20], is first investigated thusly. Specifically, all the LCEs were uniaxially stretched and annealed at 120 °C for different programming times without UV irradiation. As

presented (Fig. 3a), each curve shows that the actuation strain increases initially and slightly decreases with programming time. The maximum strains are located at 20, 15, and 10 min for the LCEs with 0.5%, 1%, and 2% PBG, respectively. The successful registration of reversible actuation verifies the homolytic bond exchange is sufficiently activated under these conditions.

To program $T_{NI}$, we now focus on determining whether the heterolytic reaction can be effectively activated after light irradiation. Therefore, the LCEs were exposed to UV light for different times first, which was conducted at 120 °C (above the $T_{NI}$) to ensure the samples were transparent. After that, they were uniaxially stretched and annealed at 120 °C for programming. Notably, the purpose here is to evaluate the contribution of UV irradiation, the programming time of the sample doped with the same amount of PBG therefore is fixed.

In order to attain the maximum actuation strain, desired for soft actuator fabrication, the programming times are 20, 15, and 10 min when the PBD amounts are 0.5%, 1%, and 2%, respectively.

The correlation between the actuation strain and irradiation time is plotted in Fig. 3b. Clearly, the strain decreases with irradiation time and eventually reaches a plateau. With ascending catalyst amount, the equilibrium time is postponed and the equilibrium strain is reduced. The more significant drop in the actuation strain (compared with Fig. 2d) reveals that the liquid crystalline content of LCEs is decreased, verifying that the TBD released after UV exposure can promote the heterolytic bond exchange. As such, the actuation strain is depressed by a longer irradiation and/or loaded with more PBG. As for the equilibrium time, it is determined by the time when TBD is fully released. A longer equilibrium time hence is required at a higher PBG dosage. Notably, the equilibrium time defines the programming window for tunning the $T_{NI}$ by photo-irradiation. The PBG loading, therefore, is fixed at 1 wt% in the subsequent investigations unless otherwise specified since it offers a considerable equilibrium strain (above 40%) and practical programming window (15 min) simultaneously.

The reduction in the liquid crystalline content was further investigated by 2D-WAXD. To conduct this, the LCEs were UV-irradiated first, followed by annealing at 120 °C for 10 min. The 2D-WAXD images of the samples with different irradiation times are presented in Supplementary Fig. 8. Clearly, all the samples exhibit the characteristic ring located at the same position, however, which becomes darker if a longer irradiation time was applied. According to the derived 1D X-ray data, the diffraction peak located around $2\theta = 5.5°$ corresponds to spacing around 16.2 Å, which is right for the end-to-end distance of rod-like molecules (Supplementary Fig. 8)[32]. This peak therefore can be attributed to the existence of LC mesogens, whose intensity is related to the remaining liquid crystalline content after programming. As such, the decrease in the peak value with ascending irradiation time verifies the liquid crystal content is reduced, which is more significant at an elevated catalyst concentration.

The $T_{NI}$s of these samples were then characterized to establish the irradiation-$T_{NI}$ paradigm. As shown in Supplementary Fig. 9, the nematic-to-isotropic transition peaks in the DSC curves are not sharp enough to quote the peak temperature as the value of the $T_{NI}$. Otherwise, the associated error is too large to allow meaningful comparison. As such, the $T_{act}$ was chosen to be investigated since it is more relevant to the actuation. As presented (Fig. 3c), the $T_{act}$ during the cooling-induced-elongation decreases with extending irradiation time, which is further summarized in Fig. 3d. Specifically, the $T_{act}$ for cooling-induced elongation falls from 105 °C to 60 °C, and the $T_{act}$ for heating-induced-contraction reduces from 135 °C to 85 °C. For more information, the correlation between the actuation strain and temperature upon heating and cooling can be found in Supplementary Fig. 10. The above results prove that the $T_{act}$ can be willingly post-programmed via varying irradiation times to regulate the degree of the network isomerization, where the network integrity is well maintained as proved by the considerable gel fractions of all programmed LCEs (>95%, Supplementary Fig. 11).

We should note that our approach of programming $T_{NI}$ via network topological isomerization has both advantages and shortcomings compared to a reported method based on tuning structural ordering[24]. On the positive side, the required programming time of our method is much shorter (minutes versus days) and the $T_{NI}$ can be defined by light, which allows much greater freedom for spatio-temporal control. As presented in Supplementary Fig. 12, the resolution attained via our method is around 0.8 mm. In comparison, the method by Yao et al. relied on continuous direct heating for days which is difficult to achieve spatial patterning of $T_{NI}$. However, our tuning of $T_{NI}$ is irreversible whereas the method by Yao et al. allows reversible adjustment of $T_{NI}$. Combining the desirable attributes of the two systems would be an interesting future direction.

## LCE actuators with complex shapes and logical actuations

As shown in Fig. 3c, both the $T_{act}$ and the actuation strain are reduced with the irradiation time, providing localized tunability by light. The actuation manners, therefore, can be diversified via introducing strain or $T_{act}$ heterogeneity into LCE networks. As presented in Fig. 3c, the actuation strain decreases from 60 to 40% due to the reduction of the mesogen content. This insignificant contrast demands to be enlarged if complex actuation heterogeneities are required. To realize this, we resort to the local actuation erasing strategy. Specifically, when a monodomain LCE is heated to a $T_P$ (>$T_{NI}$) without external stress, the bond exchange reaction is enabled to isotopically distribute the mesogens. The isotropic state consequently is fixed after cooling, which leads to the erasure of actuation strain. To confirm this, an LCE network with 60% actuation strain was heated to 120 °C and annealed for different durations. As shown in Supplementary Fig. 13, the reversible actuation is almost removed after 7 h.

From the mechanism perspective, the essence of the actuation erasing is a thermodynamically driven network rearrangement, which shall be promoted by the light-released TBD[28,29]. UV exposure was applied during the actuation erasing on that accounting. As shown in Fig. 3e, the reversible strain decreases from 60% to 0% only after 60 seconds, which is further visualized in Fig. 3f. This efficient actuation erasing is the combined effect of the homolytic and heterolytic bond exchanges after light irradiation. Thus, the heterogeneity of the actuation strain can be readily encoded via photo-mask-mediated irradiation. For simplicity, two states (60% and 0%) defined by light irradiation of 0 and 2 min are selected, corresponding to the white and yellow regions in Fig. 3g, respectively. As can be seen, planar LCE films with linear contraction/elongation can be transformed into three-dimensional (3D) structures with more complex shape-shifting actions, such as reversible volution and bending. Combined with the kirigami strategy, more sophisticated actuators like an active pyramid and a beetle are also smoothly obtained.

As for $T_{act}$, it can be tuned from 105 to 60 °C by light irradiation. Likewise, two $T_{act}$s (105 and 60 °C) defined by light irradiation of 0 and 2 min, corresponding to the white and yellow regions in Fig. 4a, were introduced into a single LCE. When heating to different temperatures, the film undergoes contraction in the different areas since the molecular order is disrupted stepwise and then progressively recovers to its initial state upon cooling. This sequential actuation then was quantitively characterized via DMA (Supplementary Fig. 14). Noteworthy, the negligible actuation observed within the UV-0 region at 80 °C is attributed to the somewhat broadening of the original $T_{NI}$ (Supplementary Fig. 3). This sort of programmable sequential actuation fashion is also readily achieved in a 3D pyramid, as presented in Supplementary Movie 1. Beyond this, when these two different $T_{act}$s were registered pixelated, a rectangular LCE film can demonstrate distinct shape changes when the temperature switches between 25, 80, and 120 °C (Supplementary Fig. 15).

The ability to spatially encode different $T_{act}$s allows for achieving actuation in a logical sequence. As presented in Fig. 4b, an LCE gripper was obtained after reversible contraction/elongation and bending were programmed into two different parts of a rectangular film (Supplementary Movie 2), in which the $T_{act}$s are also 60 and 105 °C, respectively. Therefore, the gripper can release the clip, retract to the destination upon heating, and then approach the target and grip it under cooling. Applications such as cargo transportation can be envisioned if repeating this sequential actuation via temperature switching.

Another intriguing demonstration is provided in Fig. 4c, d. Two four-leaf clovers were fabricated from a pristine and a programmed LCE for comparison, in which single and quadra-$T_{act}$s were respectively registered. Upon heating, the former one changes its original close state to an open state. However, the petals interfere with each other during cooling since they undergo reversible bending at the identical

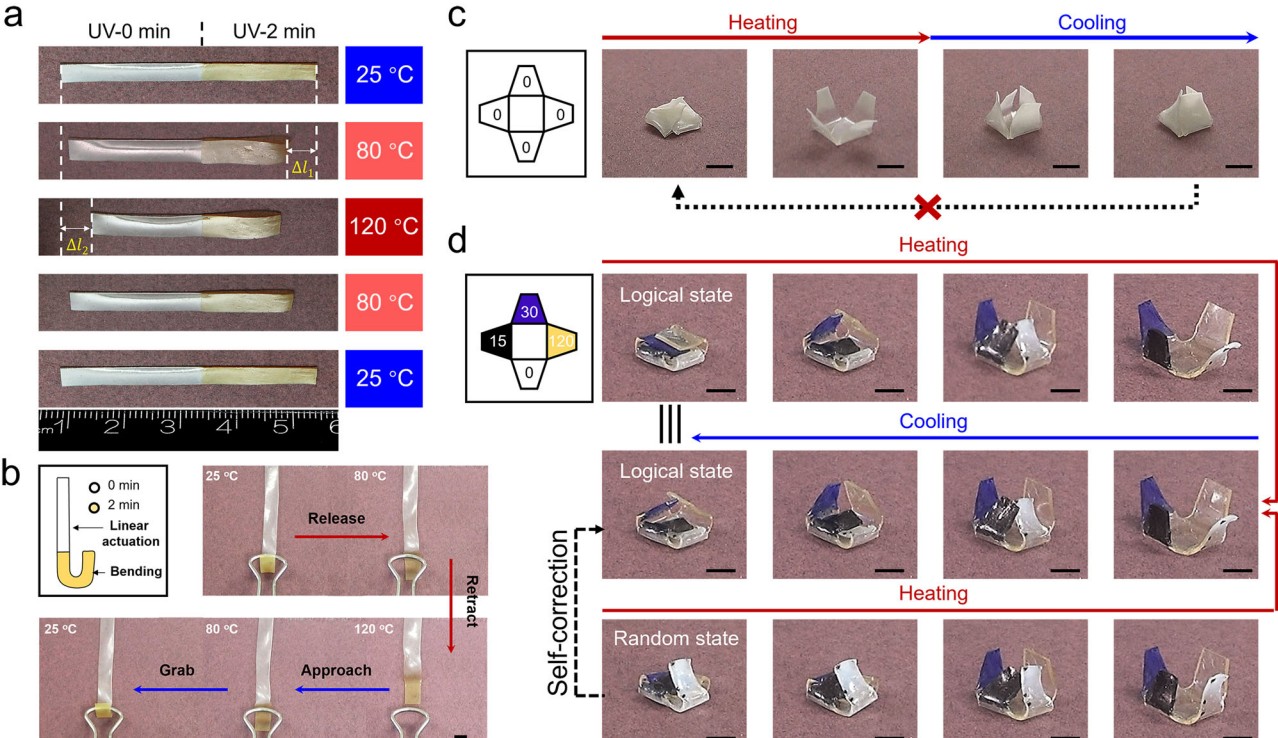

**Fig. 4 | Complex actuation manners of LCEs. a** Active LCE strip with two distinct $T_{NI}$ patterned via UV exposure exhibits a sequential actuation. **b** Reversible multi-shape actuation. The white areas are exposed to UV light for 2 min. Square size: 1 mm × 1 mm. **c**, **d** Self-locked and self-correctable actuation behavior of the four-leaf clovers with single and quadri $T_{NI}$s. The number in each petal indicates the irradiation time (s). Scale bar: 1 cm.

$T_{act}$. The recovery to its close state is blocked (Fig. 4c, Supplementary Movie 3). Comparatively, the clover based on the programmed LCE is much smarter due to the logic sequence. As shown in Fig. 4d, four different $T_{act}$s are individually encoded in each petal. Starting from a particular close state, in which the petals are stacked from top to bottom in order of $T_{act}$ (from low to high), the open state is reached after four petals are progressively actuated. Unlike the clover possessing a single $T_{act}$, it smoothly returns to the original close state since the petals recover one by one rather than simultaneously (Supplementary Movie 4). On another front, if the petals are stacked randomly in the original close state, they interfere with each other during heating since the sequence of the bending actuation is not logical. Nonetheless, this undesired close state can be self-corrected after heating above the highest $T_{act}$ and cooling to room temperature.

We note here that actuators fabricated via assembling LCE samples possessing different $T_{NI}$s together can also demonstrate the similar sequential actuations[33,34]. However, the bonding strength shall be strong enough to tolerate the shape-changing during cyclic actuations. Otherwise, delamination or disintegration occurs, which leads to the failure of actuators. In contrast, the issue never takes place in our strategy since the different $T_{NI}$s originate from the spatially distinct network topologies when the network integrity is always maintained. In the meantime, this sort of manually created heterogeneity in $T_{NI}$s is typically simple due to the limitation in hand operation. Moreover, the $T_{NI}$s of the integrated structures are not (re)programmable after assembly. As for our approach, the heterogeneity between $T_{NI}$s is arbitrarily definable with photomask-mediated irradiation, which is repeatedly tunable whenever wanted. Different requirements in practical applications hereafter can be satisfied, indicating a higher functional adaptivity. As previously mentioned, the resolution of the domain possessing different $T_{NI}$s is around 0.8 mm, which is desired for the miniaturization of LCE actuators for highly integrated soft robotic systems.

## Discussion

In summary, this work discovers a facile strategy to program the localized actuation strain and temperature for a dynamic liquid crystalline network containing aliphatic and aromatic esters in the amorphous and mesogenic phases. The homolytic bond exchange of the aliphatic esters is utilized to program the mesogen alignment. In contrast, the heterolytic bond exchange between the aliphatic and aromatic esters is invited to tune the $T_{NI}$. These two transesterification mechanisms can be distinctively activated depending on the programming conditions. Thus, the original $T_{NI}$ (115 °C) can be perseveres or lowered (to 51 °C) on-demand based on a rational selection in programming temperature or catalyst amount during the alignment programming. Using the photo-base generator, the above process can be spatial-temporally conducted. A single LCE network hence can be programmed into numerous entities with spatially distinct $T_{NI}$s, offering a handy way to realize logical actuation. The simplicity in the actuation programming (strain and temperature) shall further shed light on diversifying the shape-shifting manners of LCEs.

## Methods

### Materials

1,4-bis-[4-(6-acryloyloxyhexyloxy)-benzoyloxy]−2-methylbenzene (RM82, purity: 98%) was obtained from Beijing Bayi Space Company. 2,2′-(Ethylene-dioxyrl)-ethanethiol (EDDT, purity: 97%), ketoprofen (purity: 98%), triazabicyclo[4.4.0]dec-5-ene (TBD, purity: 98%), acetic acid (purity: 99.5%), and 2,2-dimethoxy-2-phenyl acetophenone (DMPA, purity: 98%) were purchased from Tokyo Chemical Industry. All the chemicals were used as received without further purification.

### Characterization

**Synthesis of the liquid crystalline elastomer (LCE).** 1.00 g RM82 and 0.27 g EDDT were dissolved in 1.00 g toluene at 80 °C, together with 1 wt% DMPA as the photo-initiator. After degassing, the mixture was

poured into a glass cell sandwiched by a PDMS spacer (0.4 mm), followed by a UV exposure (Uvitron Intelliray 600, 66 mW/cm$^2$, 265–700 nm) for 3 min. The obtained film was vacuum-dried in a 70 °C oven for 10 h.

**Preparation and introduction of the catalyst for transesterification.** Two catalysts were employed: neutralized TBD and photo-base generator (PBG), respectively. Specifically, the TBD was neutralized according to reference 28: 1.00 g TBD and 0.86 g acetic acid were dissolved in 3.14 g toluene and obtained a transparent solution after stirring for 1 hour. And the preparation of PBG was guided by reference 32: 0.28 g TBD, 0.52 g ketoprofen, and 3.20 g toluene were homogenously mixed, obtaining a PBG solution after stirring for 1 h. As for introducing the catalysts, the LCE film was swollen in the above solution at specific concertation (diluted by toluene) for 1 h, and the solvent was removed at room temperature in a vacuum oven for 24 h.

**Programming of the LCEs.** After the transesterification catalysts (TBD and PBG) were doped into the as-synthesized LCEs via the swelling-evaporation method, they were programmed based on an iso-strain stress relaxation approach, which was conducted via dynamic mechanical analyzer (DMA, TA Q800, Mode: stress-relaxation). Specifically, as for the LCEs doped with TBD, the samples were first heated to the targeted programming temperatures and then uniaxially stretched to a specific programming strain. Via maintaining the applied strain, the samples were annealed for different programming times. Programmed LCEs were subsequently obtained after removing the external force. As for the LCEs doped with PBG, they were first irradiated by an UV lamp (Uvitron Intelliray 600, 66 mW/cm$^2$, 265–700 nm) for different irradiation times and then programmed based on the iso-strain relaxation method. Photomasks were employed to realize patterned UV irradiations if required.

**Actuation characterization of the LCEs.** The actuation strain of the programmed sample was calculated as follows: actuation strain = $(L_{cold} - L_{heat})/L_{heat} \times 100\%$, where $L_{cold}$ and $L_{heat}$ are the sample lengths at −20 °C and 120 °C, respectively. The correlation between actuation strain and the temperature was monitored by dynamic mechanical analyzer (DMA, TA Q800) under control force mode with a negligible applied force (0.01 N). The temperature ramping rate was fixed at 2 °C/min. We note here all the presented data with error bar are calculated based on 5 tests.

**$T_{NI}$ characterization of the LCEs.** The nematic-to-isotropic temperature ($T_{NI}$) of LCEs was measured by differential scanning calorimetry (DSC, TA Q200). The temperature was scanned from −60 °C to 160 °C with a ramping rate of 5 °C/min under nitrogen.

**Wide-angle-X-ray diffraction (WAXD).** The X-ray diffraction pattern was collected via Bruker (D8, Discover) X-ray diffractometer (Cu anode, wavelength = 1.54 Å). The source parameters were set as 40 kV and 40 mA, respectively. The scanning time of each sample was fixed at 300 s.

## Data availability
Data is available from the authors upon request.

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

## Acknowledgements
This work was supported by the following programs: National Key R&D Program of China (No. 2022YFB3805701) and National Natural Science Foundation of China (No. 52203192, No. 52273112, and No. 52033009). We thank Mrs. Li Xu for the assistance in performing DSC at the State Key Laboratory of Chemical Engineering (Zhejiang University). We also thank Prof. Heiney Paul A. (University of Pennsylvania) for his constructive comments on the X-ray characterization.

## Author contributions
Q.Z., B.J., and G.C. conceived the concept and directed the project. B.J. and G.C. designed the experiments. G.C., H.F., Z.X., and Z.K. conducted the experiments. F.G. conducted the X-ray characterization. B.J., H.Y., and T.X. wrote and revised the paper. All authors analyzed and interpreted data.

## Competing interests
The authors declare no competing interests.
