## [Peer Review File · Nature Communications]

Programming actuation onset of a liquid crystalline elastomer via isomerization of network topologyREVIEWER COMMENTS

Reviewer #1 (Remarks to the Author):

Guancong Chen and coauthors present “Programming actuation onset of a liquid crystalline elastomer via isomerization of network topology.” The concepts discussed in this paper are new to the field of LCEs and the significance of this work is the ability to tune the actuation onset without dependence on the formulation itself. This work is impactful in the field in utilizing TIN (topological isomerization network) through a transesterification reaction to vary nematic to isotropic temperature of networks after polymerization and appears to be the first publication of such work. Previous works have been adequately acknowledged that address ways to alter the actuation behavior with molecular engineering, which differs from the goals of this paper. The work presented here is unique in that the effort uses dynamic chemistry to vary the topology of a polymerized LCE to control the T_{ni} . Overall, the work is well explained but the following minor revisions and clarifications are requested prior to publication.

In line 87, the LCE is said to have been made with 1,3-propane dithiol which is not the same molecule as EDDT shown in Figure 1a. DMPA is also not listed in the main text and its purpose is not explained even though it is shown in Figure 1a as part of the synthesis. Figure 1c should be clearer in which bonds are being exchanged. It is not well shown in Figure 1 how the bond exchange occurs between aliphatic and aromatic esters. The heterolytic bond exchange specifically is not well shown. How are you spatially controlling the amount of homolytic vs heterolytic bond exchange? Figure 1d is unclear with indicating what temperature each image is at currently and is difficult to follow overall. The alignment method is slightly unclear and should be further explained starting with line 118. Figure 1 indicates the reaction is a Thiol Michael addition but has no mention of a catalyst and appears to be UV activated instead so the reaction mechanism should be clarified. If it is a two-step chain extension oligomerization then photopolymerization, it should be indicated. Also, the alignment method (stretching) should be illustrated better in Figure 1. A major concern overall is the maintenance of the liquid crystallinity and general nematic phase through the bond exchange.

This might be in the SI, but if not, the POM images of the aligned domains both at 0 and 45 degrees should be shown to confirm alignment as there is some skepticism about maintaining the liquid crystallinity throughout the material. X-ray experiments have been done and show alignment, but macroscopic alignment is not shown with POM.

The network robustness is a concern after being subject to the transesterification. Gel fractions would help to show how much of it became “uncrosslinked.”

The actuation results are well presented but it is unclear as to how the T_{ni} is programmed by activating the two different transesterification mechanisms. The programming step should be more clearly explained in order to clarify this. When using the PBG, it should be better explained how the catalyst dosage is essentially spatially distributed in the material. Is there a correlation between the fact that the maximum strains are located at the same times as the programming times for each PBG sample (Line

192 and 199)?

It is mentioned that the T_{ni} in the DSC curves is insignificant for comparison, but it is not well explained. Figure 2 DSC curves do not indicate the direction of heat flow which should be included. Also, the statement between line 226 and 230 should be further explained as it is confusing as is. The authors should elaborate on what the actuation erasing strategy means. What makes the two regions described in Line 239 as yellow and white?

The issue of delamination is seemingly brushed over and not thoroughly addressed in Line 286. This should be further explained for how this occurs and how to potentially prevent it.

The methods section is sufficiently explained overall and seemingly allows for reproduction of the results. However, the programming of the LCE film is slightly confusing and further detail should be provided as to the conditions of the alignment including at what point in the LCE synthesis process that the programming is completed.

Also, a few typos and minor concerns were noticed which should be addressed as follows:

- Line 41 with the period in the superscript
- Line 64 and others, be consistent with tense of LCE vs LCEs
- TBD should be defined earlier (line 118 first appearance)
- Line 195 should say Figure 2d instead of 2d
- Line 215 missing a period

Reviewer #2 (Remarks to the Author):

The paper by Chen et al. presents a new method to tune the LCE actuation temperature using a topology isomerizable network of LCE containing aromatic and aliphatic esters in the mesogenic and amorphous phase. The transesterification among aliphatic esters can fix the alignment while the transesterification between aliphatic ester and aromatic results in the change of the TNI. Usually, the change of TNI requires the fabrication of a new material. The method here is quite facile. Before the publication, there are some major issues that the authors need to address:

1. Is there any further evidence on the transesterification? I am wondering why transesterification occurs without the presence of any hydroxyl groups.
2. Will the retro-Michael addition occur at high temperatures? Could this reaction instead of the transesterification reaction be the reason for the topology change? More solid evidence is necessary to identify the real dynamic reaction in this material.

3. When the topology changes, the micro-structure of the LCE may also change. Please at least using XRD to identify this change as it is directly related to the value of T_{ni} .

Reviewer #3 (Remarks to the Author):

Chen et al. describe a liquid crystal actuator system whereby the network can be modified post polymerisation via a bond exchange mechanism between ester groups. Two possible mechanisms are demonstrated to occur, with the heterolytic exchange enabling a tuneable decrease in the nematic-to-isotropic transition temperature (T_{ni}). Although this lowering of T_{ni} is irreversible, this paper demonstrates a novel control mechanism of actuation temperature for smart actuators. A similar mechanism was very recently described by Yao et al (DOI: 10.1038/s41467-023-39238-2) and published in June 2023. We believe this paper does not undermine the novelty of the work being reviewed here, however during the revision process, the authors here should discuss their results against the work of Yao et al.

This was an interesting paper to review and is a valuable contribution to the field. However, some revisions are required. In particular, some parts of this paper were difficult to follow, especially for those not familiar to the LCEs field, or who are not familiar with polymer chemistry and/or bond exchange mechanisms.

I recommend publication following minor corrections to address the following issues:

Main issues

1) As mentioned above, the authors should discuss their work against that of Yao et al. That is, modify their introduction to include reference to Yao et al., and also discuss their results against those of Yao et al.

2) Line 43 – The authors should provide more justification as to why manipulation of T_{ni} (or in general actuation temperature) is important. This is the core motivation for the paper, but no relevant literature is cited, or no examples are given for why this is truly important. Providing this context would aid conveying the importance of this paper.

3) Explanation of mechanisms for bond exchange. For those not familiar with LCEs and/or bond exchange mechanisms (i.e., the generalist reader of nature communications), the authors should improve their descriptions of the homolytic and heterolytic exchange processes. Figure 1c is not clear enough, in particular for the heterolytic illustration which does not demonstrate the cleaving of a mesogenic core (key to understanding the loss of liquid crystallinity). To me, the diagram implies a loss of the whole core group from the network. Following revision of figure 1c, the associated text in lines 93-95 should also be modified as appropriate to aid understanding.

4) References. I have found two reasonably significant errors with the referencing in this paper (below). Given these two errors, I kindly ask that the authors carefully check their references again prior to resubmission.

a. Line 119 - I think this reference is wrong. Ref 7 does not have anything to do with transesterification. I think the authors may have meant to reference Ref 8 here, however I do not believe Ref 8 talks about using neutralised TBD. Ideally, the authors should add a short note to aid (the non-chemists such as myself) to understand why they used a neutralised version of TBD.

b. Methods line 325, the authors refer to reference 18 for information about the use of neutralised TBD and preparing PBG. However, reference 18 does not use PBG or TBD.

Minor issues

5) Figure 1d and text in lines 104-107. The various temperatures discussed T_{low} , T_{middle} , T_{high} , T_{Nlow} , T_{Nhigh} are quite confusing. I recommend reviewing how these sentences are presented to simplify this discussion, ideally to reduce the number of T_{xx} s discussed. This would also help in a more general sense since there are numerous other temperatures mentioned and discussed throughout the paper.

6) Line 64 “adopting the concept of TIN into LCEs is a” – typo, LCEs as plural.

7) Fig2 – suggestion – be consistent with colouring of different colours for different things. For fig 2a and d, 120 and 140°C are red and blue respectively, for e and f, the colours are swapped.

8) Fig 2c caption – missed off “programming” for programming time as the authors have written elsewhere.

9) Line 118 – On its first use please write out what the acronym TBD stands for.

10) Line 119 – Please specify the (typical) thickness of the LCE films used in this work. This is relevant to potential considerations of thermal gradients present during programming times. From the pictures one might infer that the films are relatively thin and that there is a small effect of thermal gradients (i.e., outer regions undergoing more exchange reactions than inner regions) but specifying the thickness would be good to aid the reader reach a conclusion on this.

11) Line 119 – In this passage of text, the authors have not specified that a stress was applied before talking about the stress being relaxed. For readability, please state that a stress was applied before saying the stress was relaxed.

12) Line 122 – This line implies that the mesogens were aligned by the relaxation process as opposed be

being aligned by the mechanical stress. From my understanding of these systems, I would think that the relaxation fixes an oriented state which was induced by mechanical stress. Please clarify that mechanical stress is what is causing the alignment of the molecules.

13) Line 127-129 - "As presented in Figure 2d, with the same pre-stretched strain (50%), each curve shows that the actuation strain first increases with time but decreases after a longer duration, which is more significant at a higher temperature" For clarity, specify programming time and programming temperature so to distinguish from non-programming times/temperatures used for actuation. In general, please review uses of "time" and "temperature" to keep it clear when you are referring to programming conditions and (non-programming) actuation conditions.

14) Line 130 - "Further comparison among the three curves reveals that the turning points are around 10 min, 5 min, and 2.5 min, corresponding to a strain of 41%, 43%, and 55% when T_{ps} are 120 °C, 140 °C, and 160 °C, respectively." The numbers are mixed up between the times and the strains/programming temperatures.

15) Line 132 – I disagree that this conclusion is "clearly" correct. Even without heterolytic bond exchange, the higher programming temperatures could feasibly erase the stress-induced alignment thus causing low or no actuation response. This is because the material's T_{ni} is 115°C and so (even with strains applied) when above T_{ni} , the liquid crystalline groups will prefer to adopt isotropic order, thus leading to the erasure of alignment. For instance, Davidson et al, (DOI 10.1002/adma.201905682) shows a bond-exchange LCE which only features homolytic-type bond exchanges, and programming above the material's T_{ni} erased the order. The only difference here is that the programming at $T > T_{ni}$ is done under strain, but as this relieves stress it could, even without heterolytic exchange, also feasibly relieve the imposed order.

16) Line 144 - Please state the value of T_{ni} for the sample following programming at 100°C programming temperature like you have done for the other temperatures such that the sentence which follows is not as vague as it presently is: "100 °C is almost identical to the original one (115 °C)".

17) Line 148 – Please clarify and add details of how samples were tested for their soft elasticity. Critically, were tensile tests performed by straining the samples at a controlled and consistent direction relative to the programming strain orientation? One order has been programmed, the materials will be mechanically anisotropic and so the tests must have factored this into account. Without more information the reader cannot accept that the "soft elasticity originating from the mesogenic structure" becomes less obvious. All load curve in figure S4 show a clear soft elastic behaviour (arguably it is clearer in the 160°C sample. I believe think the authors are trying to distinguish between monodomain soft elasticity and polydomain soft elasticity, however more information is needed to understand what the authors tested and are presenting/arguing.

18) Line 152 - Figure 2f does not plot "the actuation strains of different samples", it plots the actuation strain normalised by the maximum actuation strain. The wording implies actuation strain vs

temperature is plotted, and with this wording the data implies all samples actuated by the same magnitude. I think the authors should make it clearer in the text that this data is used solely for determining the actuation onset strain (unless I am mistaken).

19) Line 156 - The authors have provided four actuation onset temperatures but only three T_{ps} . I assume one of these is onset actuation temperatures is for a T_p of 100°C. Please can the authors check and clarify. I am not convinced that the difference between T_{acts} and T_{nis} are trivial as they are substantial in magnitude (25°C, 8°C, 0°C, -7°C) and are not consistently different. I understand there are differences from temperatures deduced from these two thermomechanical analysis techniques, but these differences are not trivial.

20) Line 169 - "Since the temperature is hardly spatially applied" Advise re-wording for clarity to something like "As spatial patterning of temperature is difficult to realize..."

21) Figure 3d - For clarity, please specify y axis to be actuation temperature.

22) Figure 3g - Please indicate the direction of the alignment/applied aligning mechanical stress for each device.

23) Line 214 What do the authors mean by the T_{ni} peak being insignificant? Is the error associated with the values too large to allow meaningful comparison? From the figure, it's not clear how the authors have measured a value of T_{ni} . Also, there is full stop after "comparison".

24) Line 234/figure 3e I think the y axis should read "reversible", not "residual" strain.

25) Line 238 - I recommend specifying white and yellow regions of the photographs as the white region of the mask illustrations are the yellow regions of the photograph and there is some chance of confusion.

26) Line 345 – Please add details of how values of T_{ni} were determined. Were they from heating or cooling runs, and are onset or peak temperatures quoted?

27) Line 348. Please also provide indicative strain rates. Stretching speed is, for such viscoelastic materials, a fairly meaningless without knowing with the typical gauge length of the samples. Also, how comparable were the gauge lengths between tested samples? Knowing this is important for interpreting figure S4 and the information in line 148. i.e., the differences could be largely caused by differences in the strain/extension rate.

Reviewer #4 (Remarks to the Author):

I co-reviewed this manuscript with one of the reviewers who provided the listed reports. This is part of

the Nature Communications initiative to facilitate training in peer review and to provide appropriate recognition for Early Career Researchers who co-review manuscripts.

Reviewer #1 (Remarks to the Author):

Guancong Chen and coauthors present “Programming actuation onset of a liquid crystalline elastomer via isomerization of network topology.” The concepts discussed in this paper are new to the field of LCEs and the significance of this work is the ability to tune the actuation onset without dependence on the formulation itself. This work is impactful in the field in utilizing TIN (topological isomerization network) through a transesterification reaction to vary nematic to isotropic temperature of networks after polymerization and appears to be the first publication of such work. Previous works have been adequately acknowledged that address ways to alter the actuation behavior with molecular engineering, which differs from the goals of this paper. The work presented here is unique in that the effort uses dynamic chemistry to vary the topology of a polymerized LCE to control the T_{ni}. Overall, the work is well explained but the following minor revisions and clarifications are requested prior to publication.

We thank the reviewer for the constructive comments for strengthening the manuscript.

1. In line 87, the LCE is said to have been made with 1,3-propane dithiol which is not the same molecule as EDDT shown in Figure 1a. DMPA is also not listed in the main text and its purpose is not explained even though it is shown in Figure 1a as part of the synthesis. Figure 1c should be clearer in which bonds are being exchanged. It is not well shown in Figure 1 how the bond exchange occurs between aliphatic and aromatic esters. The heterolytic bond exchange specifically is not well shown. How are you spatially controlling the amount of homolytic vs heterolytic bond exchange? Figure 1d is unclear with indicating what temperature each image is at currently and is difficult to follow overall. The alignment method is slightly unclear and should be further explained starting with line 118. Figure 1 indicates the reaction is a Thiol Michael addition but has no mention of a catalyst and appears to be UV activated instead so the reaction mechanism should be clarified. If it is a two-step chain extension oligomerization then photopolymerization, it should be indicated. Also, the alignment method (stretching) should be illustrated better in Figure 1. A major concern overall is the maintenance of the liquid crystallinity and general nematic phase through the bond exchange.

Response: Thank the reviewer for these kind reminders and constructive suggestions. For clarity, the mentioned issues are divided into the following 7 parts and the corresponding responses are provided one by one:

- 1) In line 87, the LCE is said to have been made with 1,3-propane dithiol which is not the same molecule as EDDT shown in Figure 1a. DMPA is also not listed in the main text and its purpose is not explained even though it is shown in Figure 1a as part of the synthesis.

Response: We have corrected the name of the dithiol in Figure 1a. As for DMPA, it is the photo-initiator for LCE synthesis. To make this clearer, we have revised the description of the LCE synthesis on page 5 as follows:

“The topology-transformable LCE network was fabricated based on the reaction between 1,4-bis-[4-(6-acryloyloxy-hexyloxy) benzoyloxy]-2-methylbenzene (RM82, LC mesogen) and 2,2'-(ethylene-dioxyrl)-ethanethiol (EDDT, chain extender), initiated by 2,2-dimethoxy-2-phenyl-acetophenone (DMPA, photo-initiator) (Figure 1a)²⁵.”

- 2) Figure 1c should be clearer in which bonds are being exchanged. It is not well shown in Figure 1 how the bond exchange occurs between aliphatic and aromatic esters. The heterolytic bond exchange specifically is not well shown.

Response: Agree. We revised Figure 1c accordingly and added the following sentences on page 5:

“Upon heating, two distinct transesterification reactions can be activated. One is the bond exchange between the aliphatic ester groups, referred to as the “homolytic reaction”. The other one is the bond exchange between the aliphatic and aromatic ester groups, referred to as the “heterolytic reaction”. As illustrated in Figure 1c, the mesogen cores are preserved during the homolytic reaction while partially altered by the heterolytic one.”

Figure 1c: Homolytic and heterolytic bond exchange reactions.

- 3) How are you spatially controlling the amount of homolytic vs heterolytic bond exchange?

Response: The spatial control of the two reactions is based on the fact that they can be distinctively activated at different conditions, that is, the activation of the heterolytic

reaction requires a higher temperature or catalyst concentration. To make this clearer, we added the following paragraph on page 6:

“In principle, activating the heterolytic reaction within an LCE network demands a higher temperature or catalyst concentration than the homolytic one²⁷. The two reactions therefore can be distinctively triggered at different conditions, which allows us to program the T_{NI} by varying the programming temperature (T_p) and/or catalyst concentration (C_{cat}). Moreover, it can be spatially conducted when locally-different T_p s and/or C_{cat} s are applied.”

- 4) Figure 1d is unclear with indicating what temperature each image is at currently and is difficult to follow overall.

Response: Agree. To improve the clarity, we revised Figure 1d (now it is presented as Figure 1f, where only the $T_{NI-high}$ and T_{NI-low} are emphasized) and added the following description on page 6:

“Together with uniaxial stretching, a monodomain LCE with two distinct T_{NIS} (left part: T_{NI-low} , right part: $T_{NI-high}$) is obtained after programming, which can exhibit a thermally-induced stepwise actuation (Figure 1f). Upon heating, the sample’s left part contracts once the temperature is above the T_{NI-low} , whereas the right part is activated until heated above the $T_{NI-high}$. Upon cooling, the right part recovers to its initial length when the temperature is between the $T_{NI-high}$ and T_{NI-low} , while the left part elongates until the temperature continuously decreases below the T_{NI-low} .”

Figure 1f: Sequential actuation upon progressive heating/cooling.

- 5) Figure 1 indicates the reaction is a Thiol Michael addition but has no mention of a catalyst and appears to be UV activated instead so the reaction mechanism should be clarified. If it is a two-step chain extension oligomerization then photopolymerization, it should be indicated.

Response: The catalyst for thiol-Michael is DMPA and the relevant mechanism is schematically illustrated in Figure S1. To make this clearer, we added the following sentences on page 5:

“Upon UV exposure, the thiol-Michael addition (between EDDT and RM82) and the homo-polymerization of acrylate groups are simultaneously initiated by the generated free radicals (Figure S1)²⁵. The overall result is that a crosslinked LCE network is obtained after UV irradiation, where the aliphatic and aromatic esters are located within the amorphous and mesogenic phases, respectively (Figure 1b).”

Figure S1: Two reactions initiated for network synthesis.

6) Also, the alignment method (stretching) should be illustrated better in Figure 1.

Response: Agree. We revised the schematic illustration for alignment programming in Figure 1d and added the following paragraph on page 5:

“Given the bond exchange nature, the alignment programming can be realized via a thermal-mechanical approach (Figure 1d)^{8,13,22}. Specifically, when a polydomain LCE is uniaxially-stretched, the mesogens are aligned along the stretching direction. Upon heating, the dynamic network topology transformations are enabled due to the activated bond exchange reactions. The stretching-induced alignment is consequently fixed after cooling down, which can be translated into reversible actuation upon temperature change.”

Figure 1d: Network reconfiguration and isomerization during alignment programming.

- 7) A major concern overall is the maintenance of the liquid crystallinity and general nematic phase through the bond exchange.

Response: The liquid crystallinity, referring to the liquid crystalline content within the network, is reduced due to the heterolytic bond exchange reaction. This can be evidenced by the change in soft elasticity of the programmed polydomain LCEs. Specifically, through the uniaxial stretching of a polydomain LCE, when the strain reaches a certain value, the stress remains constant or even decreases when the strain continuously increases. As presented by the stress-strain curve of the sample programmed at 100 °C in Figure S4, the stress decreases from 0.58 MPa to 0.50 MPa, whereas the strain increases from 25% to 55%. This phenomenon is attributed to the reorientation of the microdomains in a polydomain LCE (see Crystals 2013, 3, 363-390, cited as reference 32). Further comparison between the four curves shows that this manner is less significant with ascending programming temperature (T_p). This is because the liquid crystalline content is considerably reduced at an elevated T_p , indicating fewer microdomains are formed. The contribution of reorientation therefore is decreased, which makes the soft elasticity less recognizable.

Figure S4: Stress-strain curves of LCEs programmed at different temperatures ($T_p = 120$ °C, $t_p = 10$ min). The tests were conducted at 25 °C with a fixed ramping rate (10 mm/min).

Additionally, the change in the liquid crystalline content can also be investigated by the 2D-X-ray Wide Angle Diffraction (2D-WAXD). To conduct this, the LCEs were UV-irradiated first, followed by annealing at 120 °C for 10 min. The 2D-WAXD images of the samples with different irradiation times are presented in Figure S11. Clearly, all the

samples exhibit the characteristic ring located at the same position, however, which becomes darker if a longer irradiation time was applied. According to the derived 1D X-ray data, the diffraction peak located around $2\theta = 5.5^\circ$ corresponds to spacing around 16.2 \AA , which is right for the end-to-end distance of rod-like molecules (Figure S9). This peak therefore can be attributed to the existence of LC mesogens, whose intensity is related to the remaining liquid crystalline content after programming (see Soft Matter, 2017, 13, 7537-7547, cited as reference 35). As such, the decrease in the peak value with ascending irradiation time verifies the liquid crystalline content is reduced, which is more significant at an elevated catalyst concentration.

Figure S9: X-ray characterization of the LCEs (a) 2D-WAXD images of LCEs with different irradiation times ($T_p = 120 \text{ }^\circ\text{C}$, $t_p = 10 \text{ min}$). (b) 1D-X-ray data derived from (a).

2. This might be in the SI, but if not, the POM images of the aligned domains both at 0 and 45 degrees should be shown to confirm alignment as there is some skepticism about maintaining the liquid crystallinity throughout the material. X-ray experiments have been done and show alignment, but macroscopic alignment is not shown with POM.

Response: Agree. We verified the mesogen alignment via polarized optical microscopy (POM). The result is provided in Figure S2 and the following discussion is added on page 8:

“The relative positions of the two arcs indicate the mesogens are aligned along the stretching direction (0° and 180°), which can be further confirmed by the polarized optical microscopy (POM) images (Figure S2).”

Figure S2: Polarized optical microscopy (POM) images of the programmed LCE (pre-stretched strain: 50%, $T_p = 120\text{ }^\circ\text{C}$, $t_p = 15\text{ min}$) under cross polarizer at (a) 0° (dark) and (b) 45° (light), respectively. Scale bar: $500\text{ }\mu\text{m}$.

3. The network robustness is a concern after being subject to the transesterification. Gel fractions would help to show how much of it became “uncrosslinked.”

Response: Agree. We investigated the gel fractions of the programmed LCEs to verify the network integrity. The results are presented in Figures S5 and S12, together with the following discussions on page 9 and 14, respectively:

For LCEs with 1% TBD (on page 9):

“The network integrity of the programmed LCEs are well maintained as proved by the considerable gel fraction ($>95\%$, Figure S5).”

Figure S5: Gel content of the LCEs programmed at different temperatures ($t_p = 10\text{ min}$). The gel content is calculated as follows: $\text{gel content (\%)} = M/M_0 \times 100\%$, where the M and M_0 are the remaining mass after swelling in toluene for 24 hours and initial mass, respectively.

For LCEs with 1% PBG (on page 14):

“The above results prove that the T_{act} can be willingly post-programmed via varying irradiation times to regulate the degree of the network isomerization, where the network integrity is well maintained as proved by the considerable gel fractions of all programmed LCEs (> 95%, Figure S12).”

Figure S12: Gel fraction of the LCEs with different irradiation times ($T_p = 120\text{ }^\circ\text{C}$, $t_p = 10\text{ min}$).

4. The actuation results are well presented but it is unclear as to how the T_{ni} is programmed by activating the two different transesterification mechanisms. The programming step should be more clearly explained in order to clarify this. When using the PBG, it should be better explained how the catalyst dosage is essentially spatially distributed in the material. Is there a correlation between the fact that the maximum strains are located at the same times as the programming times for each PBG sample (Line 192 and 199)?

Response: We appreciate these constructive suggestions.

Firstly, the programming of T_{NI} is realized via network isomerization enabled by the heterolytic reaction, which reduces the liquid crystalline content. The T_{NI} consequently is lowered depending on the isomerization degree. To make this clearer, we added the following paragraph on page 6:

“In principle, activating the heterolytic reaction within an LCE network demands a higher temperature or catalyst concentration than the homolytic one²⁷. The two reactions therefore can be distinctively triggered at different conditions, which allows us to

program the T_{NI} by varying the programming temperature (T_p) and/or catalyst concentration (C_{cat}).”

As for the photo-base generator (PBG), it can release the catalytic species (TBD) upon UV irradiation, whose amount depends on the irradiation time (See Figure R1, adapted from Sci. Adv. 2020, 6, eaaz2362). The dosage of TBD therefore can be spatially regulated by localized UV irradiation. To make this clearer, we added the schematic illustration of the reaction in Figure S8 and added the following sentence on page 11:

“Specifically, TBD, the transesterification catalyst, is released from PBG upon UV irradiation (Figure S8), whose amount depends on the irradiation time³⁴.”

Figure R1: Dependence of TBD yield on the irradiation time

Figure S8: The release of TBD under UV irradiation.

As for the programming time for each UV irradiated sample, the target is to attain the maximum actuation strain since it would be appreciated in the fabrication of soft actuators. To emphasize this, we added the following sentence on page 13:

“In order to attain the maximum actuation strain, desired for soft actuator fabrication, the programming times are 20 min, 15 min, and 10 min when the PBD amounts are 0.5%, 1%, and 2%, respectively.”

5. It is mentioned that the T_{NI} in the DSC curves is insignificant for comparison, but it is not well explained. Figure 2 DSC curves do not indicate the direction of heat flow which should be included. Also, the statement between line 226 and 230 should be further explained as it is confusing as is. The authors should elaborate on what the actuation erasing strategy means. What makes the two regions described in Line 239 as yellow and white?

Response: Thank the reviewer for the constructive suggestions.

Firstly, we claim that the T_{NI} in the DSC curves is insignificant for comparison because their transition peaks in the DSC curves are too broad to precisely quote the peak temperature as the value of the T_{NI} (Figure S10). Otherwise, the associated error is too large to allow meaningful comparison. Therefore, the actuation temperature is selected to investigate the relationship between actuation onset and irradiation time. To make this clearer, we added the following sentence on page 14:

“As shown in Figure S10, the nematic-to-isotropic transition peaks in the DSC curves are not sharp enough to quote the peak temperature as the value of the T_{NI} . Otherwise, the associated error is too large to allow meaningful comparison.”

Figure S10: DSC curves of the programmed LCEs with different irradiation times ($T_p = 120$ °C, $t_p = 10$ min).

As for Figure 2e, an arrow, together with “exo up”, has been added to indicate the direction of heat flow.

Figure 2e: T_{NIS} of the LCEs programmed at different T_{pS} (t_p : 10 min).

As for the actuation erasing strategy, it is realized via activating the bond exchange reactions when the LCEs are at the isotropic state. To make this clearer, we revised the discussion about the strain erasing strategy on page 15 as follows:

“Specifically, when a monodomain LCE is heated to a T_P ($>T_{NI}$) without external stress, the bond exchange reaction is enabled to isotopically distribute the mesogens. The isotropic state consequently is fixed after cooling, which leads to the erasure of actuation strain.”

As for the color change of the sample (from white to yellow), this is attributed to chemical reaction of PBG. As presented in Figure R2, the transparent 1% TBD solution became yellowish after irradiated with UV light for 30 s. The color change presumably originates from the aromatic product after reaction (see Figure S10 and the Response to Comment #4).

Figure R2: The color change of TBD solution after irradiated with UV for 30 s.

6. The issue of delamination is seemingly brushed over and not thoroughly addressed in Line 286. This should be further explained for how this occurs and how to potentially prevent it.

Response: We appreciate this suggestion. We mentioned the issue of delamination for emphasizing the well-maintained network integrity through our programming strategy.

To make this clearer, we added to following sentence on page 18:

“We note here that actuators fabricated via assembling LCE samples possessing different T_{NIS} together can also demonstrate the similar sequential actuations^{36,37}. However, the bonding strength shall be strong enough to tolerate the shape-changing during cyclic actuations. Otherwise, delamination or disintegration occurs, which leads to the failure of actuators. In contrast, the issue never takes place in our strategy since the different T_{NIS} originate from the spatially-distinct network topologies when the network integrity is always maintained.”

7. The methods section is sufficiently explained overall and seemingly allows for reproduction of the results. However, the programming of the LCE film is slightly confusing and further detail should be provided as to the conditions of the alignment including at what point in the LCE synthesis process that the programming is completed.

Response: Thank the reviewer for this kind suggestion. To provide a clearer picture about the LCE programming, the “Programming of the LCEs” section has been revised as follows:

“Programming of the LCEs: After the transesterification catalysts (TBD and PBG) were doped into the as-synthesized LCEs via the swelling-evaporation method, they were programmed based on an iso-strain stress relaxation approach, which was conducted via dynamic mechanical analyzer (DMA, TA Q800, Mode: stress-relaxation). Specifically, as for the LCEs doped with TBD, the samples were first heated to the targeted programming temperatures and then uniaxially-stretched to a specific programming strain. Via maintaining the applied strain, the samples were annealed for different programming times. Programmed LCEs were subsequently obtained after removing the external force. As for the LCEs doped with PBG, they were first irradiated by an UV lamp (Uvitron Intelliray 600, 66 mW/cm², 265-700 nm) for different irradiation times and then programmed based on the iso-strain relaxation method. Photomasks were employed to realize patterned UV irradiations if required.”

8. Also, a few typos and minor concerns were noticed which should be addressed as follows:

Line 41 with the period in the superscript

Line 64 and others, be consistent with tense of LCE vs LCEs

TBD should be defined earlier (line 118 first appearance)

Line 195 should say Figure 2d instead of 2d

Line 215 missing a period

Response: Thanks for the careful reading. We have corrected these typos and provided the definition of TBD before its acronym on page 7:

“To validate the hypothesized mechanism in Figure 1c, triazabicyclo[4.4.0]dec-5-ene (TBD) neutralized with acetic acid (1%) was introduced into the LCE networks as the transesterification catalyst⁷.”

Response:

Reviewer #2 (Remarks to the Author):

The paper by Chen et al. presents a new method to tune the LCE actuation temperature using a topology isomerizable network of LCE containing aromatic and aliphatic esters in the mesogenic and amorphous phase. The transesterification among aliphatic esters can fix the alignment while the transesterification between aliphatic ester and aromatic results in the change of the T_{NI} . Usually, the change of T_{NI} requires the fabrication of a new material. The method here is quite facile. Before the publication, there are some major issues that the authors need to address:

1. Is there any further evidence on the transesterification? I am wondering why transesterification occurs without the presence of any hydroxyl groups.

Response: Yes, no extra hydroxyl groups are deliberately introduced in our material design. Nevertheless, the neutralized TBD is a hygroscopic species. Its bonded water molecules can result in the hydrolysis of the ester bonds, which generates the hydroxy groups for

transesterification. (See *Polym. Chem.*, 2020, 11, 1369-1374, which is cited as reference 29). To make this clearer, we added the following sentence on page 7:

“We noted that the bonded water molecules in this hygroscopic catalyst can induce the hydrolysis of the ester bonds, generating the hydroxyl groups for transesterification²⁹.”

2. Will the retro-Michael addition occur at high temperatures? Could this reaction instead of the transesterification reaction be the reason for the topology change? More solid evidence is necessary to identify the real dynamic reaction in this material.

Response: We appreciate this suggestion. To identify the real dynamic reaction that enables the network topology transformations, a model polymer network was prepared (Figure R3). Specifically, we replaced RM82 with Bisacrylamide (BAAM) to conduct the thiol-Michael addition. As such, a polymer network without ester groups (referred to as M-network) can be obtained. After 1% TBD was doped both into the M-network and LCE network, they were subjected with the iso-strain stress relaxation test at 120 °C. The time evolution of normalized stress was presented in Figure R3b. Clearly, the stress applied to the M-network was mostly preserved while the one of the LCE network was significantly relaxed. This difference indicates that the retro-Michael addition is almost inhibited under the conditions for LCE programming. As such, we believe that the dynamic ability of our LCE networks can be attributed to the dynamic bond exchange reaction of ester bonds (transesterification).

Figure R3: Synthesis and stress relaxation of the model network. a) Synthesis route. b) Iso-strain stress relaxation tests.

3. When the topology changes, the micro-structure of the LCE may also change. Please at least using XRD to identify this change as it is directly related to the value of TNI.

Response: Agree. We investigated the micro-structure change of the programmed LCE via 2D-WAXD. The results were presented in Figure S11 and we also added the following paragraph on page 14:

“The reduction in the liquid crystalline content was further investigated by 2D-WAXD. To conduct this, the LCEs were UV-irradiated first, followed by annealing at 120 °C for 10 min. The 2D-WAXD images of the samples with different irradiation times are presented in Figure S11. Clearly, all the samples exhibit the characteristic ring located at the same position, however, which becomes darker if a longer irradiation time was applied. According to the derived 1D X-ray data, the diffraction peak located around $2\theta = 5.5^\circ$ corresponds to spacing around 16.2 Å, which is right for the end-to-end distance of rod-like molecules (Figure S9). This peak therefore can be attributed to the existence of LC mesogens, whose intensity is related to the remaining liquid crystalline content after programming³⁵. As such, the decrease in the peak value with ascending irradiation time verifies the liquid crystalline content is reduced, which is more significant at an elevated catalyst concentration.”

Figure S9: X-ray characterization of the LCEs (a) 2D-WAXD images of LCEs with different irradiation times ($T_p = 120$ °C, $t_p = 10$ min). (b) 1D-X-ray data derived from (a).

Reviewer #3 (Remarks to the Author):

Chen et al. describe a liquid crystal actuator system whereby the network can be modified post polymerization via a bond exchange mechanism between ester groups. Two possible mechanisms are demonstrated to occur, with the heterolytic exchange enabling a tunable decrease in the nematic-to-isotropic transition temperature (T_{ni}). Although this lowering of T_{ni} is irreversible, this paper demonstrates a novel control mechanism of actuation temperature for smart actuators. A similar mechanism was very recently described by Yao et al (DOI: 10.1038/s41467-023-39238-2) and published in June 2023. We believe this paper does not undermine the novelty of the work being reviewed here, however during the revision process, the authors here should discuss their results against the work of Yao et al.

This was an interesting paper to review and is a valuable contribution to the field. However, some revisions are required. In particular, some parts of this paper were difficult to follow, especially for those not familiar to the LCEs field, or who are not familiar with polymer chemistry and/or bond exchange mechanisms.

I recommend publication following minor corrections to address the following issues:

We thank the reviewer for the positive comments and specific suggestions below that help us strengthen the paper.

Main issues

1) As mentioned above, the authors should discuss their work against that of Yao et al. That is, modify their introduction to include reference to Yao et al., and also discuss their results against those of Yao et al.

Response: We appreciate this suggestion. As the reviewer suggested, we added the following discussion in the Introduction section (on page 3) :

“We should note that Yao et al. recently reported that annealing a dynamic covalent LCE network can alter the T_{NI} , which was attributed to the change in the structural order of the mesogenic phase. Specifically, annealing at $T_a < T_{NI}$ encourages forming a more compact layered structure corresponding to a higher structural order and higher T_{NI} . In contrast, annealing at $T_a > T_{NI}$ had an opposite effect, that is, it impaired the regularity of the lamellar stacking and led to a lower T_{NI} . The change in structural order and T_{NI} can be fixed by quenching the bond exchange reaction. Despite the elegance, this post-modification of T_{NI} required a long annealing time (usually days) and the chemical topologies was not altered in the process.”

Meanwhile, we have added the following paragraph on page 14 to emphasize the difference between our approach and the annealing strategy demonstrated by Yao et al.:

“We should note that our approach of programming T_{NI} via network topological isomerization has both advantages and shortcomings compared to a reported method based on tuning structural ordering. On the positive side, the required programming time of our method is much shorter (minutes versus days) and the T_{NI} can be defined by light, which allows much greater freedom for spatio-temporal control. As presented in Figure S13, the resolution attained via our method is around 0.8 mm. In comparison, the method by Yao et al. relied on continuous direct heating for days which is difficult to achieve spatial patterning of T_{NI} . However, our tuning of T_{NI} is irreversible whereas the method by Yao et al. allows reversible adjustment of T_{NI} . Combining the desirable attributes of the two systems would be an interesting future direction.”

2) Line 43 – The authors should provide more justification as to why manipulation of T_{ni} (or in general actuation temperature) is important. This is the core motivation for the paper, but no relevant literature is cited, or no examples are given for why this is truly important. Providing this context would aid conveying the importance of this paper.

Response: We appreciate this constructive suggestion. To emphasize the importance of programming the T_{NI} , we added to the following sentences on page 2:

“Mechanistically speaking, the former indicates the attainable functions of a soft actuator and the latter defines its applicable application scenarios. For instance, the T_{NI} of LCEs for

biomedical applications typically shall be lowered to 70 °C⁴. Manipulating the T_{NI} hereafter paves the way to make LCEs functional in distinct conditions. Beyond this, sequential actuation, indicating higher controllability in shape-changing, can be realized via encoding multiple T_{NIs} into one LCE, which would allow us to explore new applications of LCE-based devices.”

3) Explanation of mechanisms for bond exchange. For those not familiar with LCEs and/or bond exchange mechanisms (i.e., the generalist reader of nature communications), the authors should improve their descriptions of the homolytic and heterolytic exchange processes. Figure 1c is not clear enough, in particular for the heterolytic illustration which does not demonstrate the cleaving of a mesogenic core (key to understanding the loss of liquid crystallinity). To me, the diagram implies a loss of the whole core group from the network. Following revision of figure 1c, the associated text in lines 93-95 should also be modified as appropriate to aid understanding.

Response: We appreciate this suggestion. The following sentences are added on page 5 to provide a detail explanation about the bond exchange mechanism, together with a revised Figure 1c:

“Upon heating, two distinct transesterification reactions can be activated. One is the bond exchange between the aliphatic esters, referred to as the “homolytic reaction”. The other one is the bond exchange reaction between the aliphatic and aromatic ester groups, referred to as the “heterolytic reaction”.”

Figure 1c: Homolytic and heterolytic bond exchange reactions.

4) References. I have found two reasonably significant errors with the referencing in this paper (below). Given these two errors, I kindly ask that the authors carefully check their references again prior to resubmission.

a. Line 119 - I think this reference is wrong. Ref 7 does not have anything to do with transesterification. I think the authors may have meant to reference Ref 8 here, however I do not

believe Ref 8 talks about using neutralised TBD. Ideally, the authors should add a short note to aid (the non-chemists such as myself) to understand why they used a neutralised version of TBD.

b. Methods line 325, the authors refer to reference 18 for information about the use of neutralised TBD and preparing PBG. However, reference 18 does not use PBG or TBD.

Response: Thank the reviewer for the careful reading. We have double checked the references to make sure they are right cited. As for the neutralization of TBD, the purpose is to suppress its alkalinity. To make this clearer, we added the following sentence on page 7:

“The neutralization of TBD is to suppress its alkalinity. Otherwise, the thiol-Michael addition is initiated by it before UV irradiation, which makes the precursor solution too viscous to be handed²⁸.”

Minor issues

5) Figure 1d and text in lines 104-107. The various temperatures discussed T_{low} , T_{middle} , T_{high} , T_{NIlow} , T_{NIhigh} are quite confusing. I recommend reviewing how these sentences are presented to simplify this discussion, ideally to reduce the number of T_{xx} s discussed. This would also help in a more general sense since there are numerous other temperatures mentioned and discussed throughout the paper.

Response: Agree. Too many temperatures are discussed in Figure 1d, which makes it difficult to follow. To improve the clarity, we have revised the Figure 1d and the description about the thermally-triggered stepwise actuation on page 6 as follows:

“Together with uniaxial stretching, a monodomain LCE with two distinct T_{NIS} (left part: T_{NI-low} , right part: $T_{NI-high}$) is obtained after programming, which can exhibit a thermally-induced stepwise actuation (Figure 1f). Upon heating, the sample’s left part contracts once the temperature is above the T_{NI-low} , whereas the right part is activated until heated above the $T_{NI-high}$. Upon cooling, the right part recovers to its initial length when the temperature is between the $T_{NI-high}$ and T_{NI-low} , while the left part elongates until the temperature continuously decreases below the T_{NI-low} .”

Figure 1f: Stepwise actuation of a LCE programmed with two T_{NIS} .

6) Line 64 “adopting the concept of TIN into LCEs is a” – typo, LCEs as plural.

Response: Thank the reviewer for the kind reminder. We have corrected this typo on page 3.

“It should be emphasized that adopting the concept of TIN into LCEs is a challenging task, as the mesogen alignment should be well maintained during the topology isomerization for reversible actuation.”

7) Fig2 – suggestion – be consistent with colouring of different colours for different things. For fig 2a and d, 120 and 140°C are red and blue respectively, for e and f, the colours are swapped.

Response: Thanks for this kind suggestion. The colours used in Figure 2 are revised to make sure they are consistent. Specifically, the black, red, blue, and green lines represent the programming temperature of 100 °C, 120 °C, 140 °C, and 160 °C, respectively.

8) Fig 2c caption – missed off “programming” for programming time as the authors have written elsewhere.

Response: Yes, we now added “programming” in the caption of Figure 2c to ensure it is consistent with others.

9) Line 118 – On its first use please write out what the acronym TBD stands for.

Response: Thanks for this kind reminder. We added the full name of TBD before its acronym on page 7 as follows:

“To validate the hypothesized mechanism in Figure 1c, triazabicyclo[4.4.0]dec-5-ene (TBD) neutralized with acetic acid (1%) was introduced into the LCE networks as the transesterification catalyst⁷.”

10) Line 119 – Please specify the (typical) thickness of the LCE films used in this work. This is relevant to potential considerations of thermal gradients present during programming times. From the pictures one might infer that the films are relatively thin and that there is a small effect of thermal gradients (i.e., outer regions undergoing more exchange reactions than inner regions) but specifying the thickness would be good to aid the reader reach a conclusion on this.

Response: Agree. As the reviewer suggested, we added the following sentence on page 7 to specify the thickness.

“The thickness of the LCE films in this paper is fixed at 0.4 mm to exclude the thermal gradient along the thickness direction.”

11) Line 119 – In this passage of text, the authors have not specified that a stress was applied before talking about the stress being relaxed. For readability, please state that a stress was applied before saying the stress was relaxed.

Response: We appreciate this suggestion. As the reviewer suggested, we revised the discussion about the stress relaxation on page 7 as follows:

“Specifically, each sample was stretched to 50% first and annealed at a specific programming temperature (T_p). The time evolution of the ratio between the residual stress (σ) and the initial applied stress (σ_0), referred to as the normalized stress (σ/σ_0), was monitored (Figure 2a).”

12) Line 122 – This line implies that the mesogens were aligned by the relaxation process as opposed to being aligned by the mechanical stress. From my understanding of these systems, I would think that the relaxation fixes an oriented state which was induced by mechanical stress. Please clarify that mechanical stress is what is causing the alignment of the molecules.

Response: Yes, the oriented state is induced by mechanical stress, which is subsequently fixed stress due to dynamic network topology transformations. To make this clearer, we added the following sentences on page 8:

“The verified network dynamicability allows us to program the mesogen alignment via the thermal-mechanical approach. Specifically, mesogens are aligned along the stretching direction when the LCE is uniaxially-stretched. The subsequently activated bond exchange upon heating can fix the alignment after cooling down.”

13) Line 127-129 - “As presented in Figure 2d, with the same pre-stretched strain (50%), each curve shows that the actuation strain first increases with time but decreases after a longer duration, which is more significant at a higher temperature” For clarity, specify programming time and programming temperature so to distinguish from non-programming times/temperatures used for actuation. In general, please review uses of “time” and “temperature” to keep it clear when you are referring to programing conditions and (non-programming) actuation conditions.

Response: Agree. As the reviewer suggested, the temperature and time for alignment programming is specified as “programming temperature” and “programming time”, respectively. We have overviewed the whole manuscript and revised them accordingly.

14) Line 130 - “Further comparison among the three curves reveals that the turning points are around 10 min, 5 min, and 2.5 min, corresponding to a strain of 41%, 43%, and 55% when T_{ps} are 120 °C, 140 °C, and 160 °C, respectively.” The numbers are mixed up between the times and the strains/programming temperatures.

Response: Thank the reviewer for this kind reminder. To make this clearer, this sentence now has been revised as follows:

“Further comparison among the three curves reveals that the turning points are around 10 min, 5 min, and 2.5 min when T_{ps} are 120 °C, 140 °C, and 160 °C, respectively. And the λ at each turning point is about 41%, 43%, and 55%.”

15) Line 132 – I disagree that this conclusion is “clearly” correct. Even without heterolytic bond exchange, the higher programming temperatures could feasibly erase the stress-induced alignment

thus causing low or no actuation response. This is because the material's T_{ni} is 115°C and so (even with strains applied) when above T_{ni} , the liquid crystalline groups will prefer to adopt isotropic order, thus leading to the erasure of alignment. For instance, Davidson et al, (DOI 10.1002/adma.201905682) shows a bond-exchange LCE which only features homolytic-type bond exchanges, and programming above the material's T_{ni} erased the order. The only difference here is that the programming at $T > T_{ni}$ is done under strain, but as this relieves stress it could, even without heterolytic exchange, also feasibly relieve the imposed order.

Response: We agree with the reviewer's comment. The decrease in actuation strain should be attributed to the fact the LCE network prefers to adopt the isotropic state when the stress is significantly relaxed. This can be proved by the fact that the actuation strain of the sample programmed at 160 °C for 10 min was almost erased while it still possessed a T_{NI} around 51 °C. If the actuation erasing was realized based on the heterolytic reaction as we claimed previously, the nematic-to-isotropic transition would be eliminated when the alignment is almost erased, which is contradictory to the DSC result (Figure 2e). To correct this, We revised the relevant discussion on page 9 as follows and cited the mentioned paper as reference 30:

“In principle, the drop in λ at the later period of programming is attributed to the fact that the applied stress is already significantly relaxed. Under such condition, the LCE network prefers to adopt an isotropic state to miniaturize the network entropy since T_p is above the T_{NI} , although the strain is maintained. The fixed alignment consequently is gradually erased and the actuation strain is reduced³⁰.”

16) Line 144 - Please state the value of T_{ni} for the sample following programming at 100°C programming temperature like you have done for the other temperatures such that the sentence which follows is not as vague as it presently is: “100 °C is almost identical to the original one (115 °C)”.

Response: Agree. The value of T_{NI} for the sample programmed at 100 °C is now specified in the following sentence on page 9.

“Notably, the T_{NI} of the LCE programmed at 100 °C is about 114 °C, which is almost identical to the original one (115 °C).”

17) Line 148 – Please clarify and add details of how samples were tested for their soft elasticity. Critically, were tensile tests performed by straining the samples at a controlled and consistent direction relative to the programming strain orientation? One order has been programmed, the materials will be mechanically anisotropic and so the tests must have factored this into account. Without more information the reader cannot accept that the “soft elasticity originating from the mesogenic structure” becomes less obvious. All load curve in figure S4 show a clear soft elastic behaviour (arguably it is clearer in the 160°C sample. I believe the authors are trying to distinguish between monodomain soft elasticity and polydomain soft elasticity, however more information is needed to understand what the authors tested and are presenting/arguing.

Response: We appreciate this suggestion. The uniaxial stretching tests were conducted with polydomain LCEs. The soft elasticity therefore can be attributed to the reorientation of the microdomains within a polydomain LCE network. To provide more information, we revised the relevant paragraph on page 9 as follows:

“To evidence this, the change in soft elasticity of polydomain LCEs after programming is monitored. Specifically, LCEs were annealed at 100 °C, 120 °C, 140 °C, and 160 °C for 10 min without external force. After cooling, they were subjected to uniaxial stretching tests at 25 °C (below T_{NI}) with a fixed strain ramping rate (10 mm/min). In principle, the soft elasticity can be noticed in the stress-strain curve. Specifically, after reaching a certain value of strain, the stress remains constant or even decreases when the strain continuously increases. This phenomenon is attributed to the reorientation of the microdomains in polydomain LCEs^{31,32}. Comparison between the four stress-strain curves in Figure S5 shows that this unique manner becomes insignificant with ascending T_p . This is because the liquid crystalline content is reduced more considerably at an elevated T_p , indicating less microdomains are formed. The contribution of reorientation therefore is decreased, which makes the soft elasticity less recognizable.”

18) Line 152 - Figure 2f does not plot “the actuation strains of different samples”, it plots the actuation strain normalised by the maximum actuation strain. The wording implies actuation strain vs temperature is plotted, and with this wording the data implies all samples actuated by the same

magnitude. I think the authors should make it clearer in the text that this data is used solely for determining the actuation onset strain (unless I am mistaken).

Response: Agree. The purpose of Figure 2f is to find the value of actuation temperature, which is defined as the temperature when the actuation strain reaches 5% of the maximum actuation strain. We therefore plotted the normalized actuation strain as the function of temperature. However, we used “actuation strain” rather “normalized actuation strain” in the main text, which might make the reviewer confused. Therefore, we revised the relevant paragraph on page 9 as follows:

“The correlation between the actuation strain (λ) and temperature is more important when considering LCEs as actuators. In particular, the temperature when the actuation starts is a pivotal parameter to portray the shape-changing behavior of LCEs, which is referred to as actuation temperature (T_{act}). For a quantitative comparison, it is defined as the temperature when the normalized actuation strain ($\lambda/\lambda_{max} \times 100\%$, λ_{max} is the total strain of one actuation cycle) reaches 5%. To investigate the relationship between T_{act} and T_p , the normalized actuation strain of the LCEs programmed at different T_p s for 10 min was plotted as a temperature function in Figure S6.”

19) Line 156 - The authors have provided four actuation onset temperatures but only three T_p s. I assume one of these is onset actuation temperatures is for a T_p of 100°C. Please can the authors check and clarify. I am not convinced that the difference between T_{acts} and T_{nis} are trivial as they are substantial in magnitude (25°C, 8°C, 0°C, -7°C) and are not consistently different. I understand there are differences from temperatures deduced from these two thermomechanical analysis techniques, but these differences are not trivial.

Response: Yes, the first T_{act} is for the sample programmed at 100 °C. The relevant sentence now has revised as follows:

“Accordingly, Figure 2f shows the T_{acts} are 78 °C, 68 °C, 52 °C, and 38 °C when the T_p s are 100 °C, 120 °C, 140 °C, and 160 °C, respectively”

We agree that the difference is not trivial and added the following paragraph on page 10:

“Further comparison between the T_{act} and T_{NI} of one LCE shows their values are not the same. For instance, the T_{act} is 68 °C whereas the T_{NI} is 82 °C when T_p is 120 °C. This difference can be attributed to the following reasons: i) The T_{NI} is determined based on the enthalpic change of a microscopic transition, while the T_{act} is deduced from the temperature evolution of the macroscopic length; ii) The T_{NI} is investigated via DSC, while the T_{act} is characterized via DMA. The intrinsic variance between the two analysis techniques needs to be also considered.”

20) Line 169 - “Since the temperature is hardly spatially applied” Advise re-wording for clarity to something like “As spatial patterning of temperature is difficult to realize...”

Response: We appreciate this suggestion. The mentioned sentence now has revised as follows:

“Nevertheless, the spatial patterning of the temperature is difficult to realize.”

21) Figure 3d - For clarity, please specify y axis to be actuation temperature.

Response: Agree. The y axis of Figure 3d now has been specified as “actuation temperature”.

Figure 3d: Dependence of the actuation temperature on irradiation time (T_p : 120 °C, t_p : 10 mins, pre-stretched strain: 50%).

22) Figure 3g - Please indicate the direction of the alignment/applied aligning mechanical stress for each device.

Response: We appreciate this constructive suggestion. A schematic illustration of each device is added to right corner of each top picture and the alignment direction is indicated by the direction of the arrows.

Figure 3g: 3D active LCE structures fabricated via spatially erasing actuation strain. Inset: the left one is the employed photomask, where the yellow areas are the exposed regions; The right one is the device scheme, where the arrows indicate the alignment direction and the solid lines represent the cutting lines of the kirigami patterns. Scale bars: 1 cm.

23) Line 214 What do the authors mean by the T_{ni} peak being insignificant? Is the error associated with the values too large to allow meaningful comparison? From the figure, it's not clear how the authors have measured a value of T_{ni} . Also, there is full stop after “comparison”.

Response: Yes, the “insignificant” means that the nematic-to-isotropic transition peaks in the DSC curves are not sharp enough, quoting the peak tempera as the value of the T_{NI} vulnerary makes the errors too large to allow meaningful comparison. To make this clearer, we have revised the relevant discussion as follows:

“As shown in Figure S10, the nematic-to-isotropic transition peaks in the DSC curves are not sharp enough to quote the peak temperature as the value of the T_{NI} . Otherwise, the associated error is too large to allow meaningful comparison.”

The measuring method of the T_{NI} is now specified on page 9 as follows:

“The curves of the second heating run were presented in Figure 2e, where the peak temperature was quoted as the value of T_{NI} .”

The full stop after “comparison” has been removed.

24) Line 234/figure 3e I think the y axis should read “reversible”, not “residual” strain.

Response: Agree. The y axis of Figure 3e now has been revised as “actuation strain”.

Figure 3e. Residual actuation strain of a monodomain LCE irradiated at 120 °C for different times (without stress).

25) Line 238 - I recommend specifying white and yellow regions of the photographs as the white region of the mask illustrations are the yellow regions of the photograph and there is some chance of confusion.

Response: Thanks for this constructive suggestion. As suggested, we have revised the schematic illustration of the employed photomasks in Figure 3d. Now, the white and yellow regions indicate the unexposed and exposed areas, respectively, which is consistent with the color change (from white to yellow) of the samples after been irradiated. The revised Figure 3d is provided in Response to Comment #22.

26) Line 345 – Please add details of how values of T_{NI} were determined. Were they from heating or cooling runs, and are onset or peak temperatures quoted?

Response: We appreciate this suggestion. The measuring method of T_{NI} can be found in Response to Comment #23.

“The curves of the second heating run were presented in Figure 2e, where the peak temperature was quoted as the value of T_{NI} .”

27) Line 348. Please also provide indicative strain rates. Stretching speed is, for such viscoelastic materials, a fairly meaningless without knowing with the typical gauge length of the samples. Also, how comparable were the gauge lengths between tested samples? Knowing this is important for interpreting figure S4 and the information in line 148. i.e., the differences could be largely caused by differences in the strain/extension rate.

Response: Agree, the strain rate and gauge length are pivotal parameters when conducting the uniaxial stretching test. We specified the strain rate on pages 9 as follows:

“After cooling, they were subjected to uniaxial stretching tests at 25 °C (below T_{NI}) with a fixed strain ramping rate (10 mm/min).”

The gauge length of test samples was specified in the Method Section as follows:

“The gauge length of the test sample was fixed at 15 mm.”

Reviewer #4 (Remarks to the Author):

Response: Thank the reviewer for the constructive comments for improving our manuscript.

REVIEWERS' COMMENTS

Reviewer #1 (Remarks to the Author):

The authors have adequately responded to my and the other reviewers queries.

Reviewer #2 (Remarks to the Author):

I am satisfied with the revision and have no further questions.

Reviewer #3 (Remarks to the Author):

I thank Chen et al. for their revised manuscript and addressing most of my concerns. I still have three outstanding points related to my original points 3, 14, and 17. I believe this paper can be published following minor corrections to address these remaining points.

Point 3)

The added figure is welcome, however I believe it still causes confusion and the opportunity for mis-interpretation, especially for the non-expert. In the diagram for the homolytic bond exchange, two exchanges are shown to have taken place with the effect of swapping the mesogenic cores and ultimately leaving the network unchanged and no relaxation of stress. Showing only one exchange (by way of example) would be better and would show that the chains have been reconfigured. Similarly, for the heterolytic mechanism, the diagram implies multiple exchanges occur simultaneously and no network relaxation. I suggest the authors should just show the example of one exchange taking place to make it clearer how the backbones are swapping through these exchange mechanisms.

Point 14)

I think the authors have still mixed up the order of the turning points for the maximum actuation strains. For 120, 140, and 160°C, the turning points are at 10, 5 and 2.5 minutes respectively, which the authors have correct, and the actuation strains are 55, 43 and 41% respectively, which the authors have listed in reverse. Please can the authors double check this again.

Point 17) I thank the author for the added details around the testing of their polydomains for soft elasticity. However, the revisions do open more questions and I am not convinced by the arguments made, and indeed I believe they are unnecessary. If the programming temperature is reducing the liquid crystallinity of the LCE, then this would reduce the order parameter and hence polymer anisotropy within each domain of the material. The effect of this would be that the length of the soft elastic plateau would be reduced, and as you approach all order being erased (as with the 160°C @ 10 minute sample) the load curve should look like that of a conventional elastomer, which can look similar to a soft elastic load curve making this type of analysis inherently difficult.

In these data the difficulties in drawing conclusions come from the following. First, these samples show a surprisingly high onset strain for polydomain soft elasticity (between about 15 and 30%). Typically, in LCEs (see book by Warner and Terentjev) there is minimal or even no detectable onset strain for the polydomain soft elastic effect. Second the data does not clearly show a reduction in the soft elastic plateau length as one would expect, and which would be difficult to record as explained above. This is also made more difficult by the change in the stress level of the plateau. I believe there these programming treatments have more effects on the load curve shape than just on the soft elasticity and so until these combined effects are better understood i.e. in a future publication, I recommend removing these load curves and the discussion around soft elasticity. The authors already have enough convincing data showing how the heterolytic bond exchange affects T_{ni} and the actuation strain. As a side note, the authors have accidentally referred to figure S5 instead of S4.

Reviewer #4 (Remarks to the Author):

Reviewer #1 (Remarks to the Author):

The authors have adequately responded to my and the other reviewers queries.

Reviewer #2 (Remarks to the Author):

I am satisfied with the revision and have no further questions.

Response: We thank Reviewers #1 and #2 for their positive comments on our revised manuscript.

Reviewer #3 (Remarks to the Author):

I thank Chen et al. for their revised manuscript and addressing most of my concerns. I still have three outstanding points related to my original points 3, 14, and 17. I believe this paper can be published following minor corrections to address these remaining points.

Response: Thank the reviewer for the constructive comments that help us strengthen our manuscript.

Point 3)

The added figure is welcome, however I believe it still causes confusion and the opportunity for mis-interpretation, especially for the non-expert. In the diagram for the homolytic bond exchange, two exchanges are shown to have taken place with the effect of swapping the mesogenic cores and ultimately leaving the network unchanged and no relaxation of stress. Showing only one exchange (by way of example) would be better and would show that the chains have been reconfigured. Similarly, for the heterolytic mechanism, the diagram implies multiple exchanges occur simultaneously and no network relaxation. I suggest the authors should just show the example of one exchange taking place to make it clearer how the backbones are swapping through these exchange mechanisms.

Response: Agree. As the reviewer suggested, Figure 1c is revised as follows:

Figure 1c: Homolytic and heterolytic bond exchange reactions.

Point 14)

I think the authors have still mixed up the order of the turning points for the maximum actuation strains. For 120, 140, and 160°C, the turning points are at 10, 5 and 2.5 minutes respectively, which the authors have correct, and the actuation strains are 55, 43 and 41% respectively, which the authors have listed in reverse. Please can the authors double check this again.

Response: Thank the reviewer for the careful reading. We have corrected the claim on page 8 as follows:

“And the λ at each turning point is about 55%, 43%, and 41%.”

Point 17) I thank the author for the added details around the testing of their polydomains for soft elasticity. However, the revisions do open more questions and I am not convinced by the arguments made, and indeed I believe they are unnecessary. If the programming temperature is reducing the liquid crystallinity of the LCE, then this would reduce the order parameter and hence polymer anisotropy within each domain of the material. The effect of this would be that the length of the soft elastic plateau would be reduced, and as you approach all order being erased (as with the 160°C @ 10 minute sample) the load curve should look like that of a conventional elastomer, which can look similar to a soft elastic load curve making this type of analysis inherently difficult.

In these data the difficulties in drawing conclusions come from the following. First, these samples show a surprisingly high onset strain for polydomain soft elasticity (between about 15 and 30%). Typically, in LCEs (see book by Warner and Terentjev) there is minimal or even no detectable onset strain for the polydomain soft elastic effect. Second the data does not clearly show a reduction in the soft elastic plateau length as one would expect, and which would be difficult to record as explained above. This is also made more difficult by the change in the stress level of the plateau. I believe there these programming treatments have more effects on the load curve shape than just on the soft elasticity and so until these combined effects are better understood i.e., in a future publication, I recommend removing these load curves and the discussion around soft elasticity. The authors already have enough convincing data showing how the heterolytic bond exchange affects T_{ni} and the actuation strain. As a side note, the authors have accidentally referred to figure S5 instead of S4.

Response: Thank the reviewer for the constructive comment. As the reviewer suggested, we removed the characterization and discussion about the soft elasticity to avoid unnecessary misleading.

Reviewer #4 (Remarks to the Author):

Response: Thank the reviewer for your kind help in our manuscript revision.